# The novel SH3 domain protein Dlish/CG10933 mediates fat signaling in Drosophila by binding and regulating Dachs

**Yifei Zhang[1†], Xing Wang[1†], Hitoshi Matakatsu[1,2], Richard Fehon[2], Seth S Blair[1*]**

[1]Department of Zoology, University of Wisconsin-Madison, Madison, United States; [2]Department of Molecular Genetics and Cell Biology, University of Chicago, Chicago, United States

**Abstract** Much of the Hippo and planar cell polarity (PCP) signaling mediated by the Drosophila protocadherin Fat depends on its ability to change the subcellular localization, levels and activity of the unconventional myosin Dachs. To better understand this process, we have performed a structure-function analysis of Dachs, and used this to identify a novel and important mediator of Fat and Dachs activities, a Dachs-binding SH3 protein we have named Dlish. We found that Dlish is regulated by Fat and Dachs, that Dlish also binds Fat and the Dachs regulator Approximated, and that Dlish is required for Dachs localization, levels and activity in both wild type and *fat* mutant tissue. Our evidence supports dual roles for Dlish. Dlish tethers Dachs to the subapical cell cortex, an effect partly mediated by the palmitoyltransferase Approximated under the control of Fat. Conversely, Dlish promotes the Fat-mediated degradation of Dachs.

*For correspondence: ssblair@wisc.edu

†These authors contributed equally to this work

Competing interests: The authors declare that no competing interests exist.

## Introduction

Heterophilic binding between the giant Drosophila protocadherins Fat and Dachsous (Ds) both limits organ growth, via regulation of the Hippo pathway, and orients planar cell polarity (PCP), through cell-by-cell polarization of Fat, Ds and their downstream effectors (*Blair, 2012*, *2014*; *Irvine and Harvey, 2015*; *Matis and Axelrod, 2013*). Loss of Fat and, to a lesser extent, Ds, leads to the profound overgrowth of the Drosophila imaginal discs that give rise to adult appendages, and loss of either disorders the polarity of cell divisions, hairs and other morphological features in a variety of Drosophila tissues. But while players and pathways have been defined that are genetically downstream of Fat-Ds binding, only a little is known about the biochemical links between these and their most powerful regulator, the intracellular domain (ICD) of Fat.

A good deal of the recent work on Fat effectors has focused on the regulation of unconventional type XX myosin Dachs (*Mao et al., 2006*; *Matakatsu and Blair, 2008*; *Rauskolb et al., 2011*; *Rogulja et al., 2008*). Dachs is critical first because it provides the only known marker specifically sensitive to changes in the Fat/Ds branches of both the Hippo and PCP pathways. Dachs is normally concentrated in the subapical cell cortex, overlapping subapically-concentrated Fat and Ds. Loss of Fat greatly increases subapical Dachs levels, and polarization of Fat and Ds to opposite cell faces can in turn polarize Dachs to the face with less Fat (*Ambegaonkar AA, 2012*; *Bosveld et al., 2012*, *2016*; *Brittle et al., 2012*; *Mao et al., 2006*). Fat thus inhibits or destabilizes subapical Dachs, while Ds may do the opposite. Downstream changes in Hippo or PCP activities do not affect Dachs.

Dachs changes are also critical because they play a major role downstream of Fat. Dachs binds to and inhibits the activity of the kinase Warts (the Drosophila Lats1/2 ortholog), both reducing Warts

levels and changing its conformation (*Cho et al., 2006*; *Rauskolb et al., 2011*; *Vrabioiu and Struhl, 2015*). Warts is concentrated in the subapical cell cortex (*Matakatsu and Blair, 2008*; *Sun et al., 2015*), and thus the increased cortical Dachs of *fat* mutants should reduce the phosphorylation of Yorkie by Warts, allowing Yorkie to move into the nucleus to drive the transcription of growth-promoting target genes. Indeed, Dachs is necessary for the overgrowth and increased Yorkie target gene expression of *fat* mutants (*Cho et al., 2006*; *Mao et al., 2006*). Dachs overexpression also causes overgrowth, although more weakly than the overgrowth caused by the loss of Fat, indicating that Dachs is partly sufficient (*Cho et al., 2006*; *Rauskolb et al., 2011*).

Dachs can also bind to the core PCP pathway component Spiny legs (Sple) and alter its localization, thus influencing PCP in the subset of tissues that rely on Sple (*Ayukawa et al., 2014*). The increased levels of unpolarized Dachs in *fat* mutants may misdirect Sple, accounting for at least some of the PCP defects; *fat* mutant hair PCP defects are improved, although not eliminated, by loss of *dachs* (*Ambegaonkar and Irvine, 2015*; *Blair, 2014*; *Mao et al., 2006*; *Matakatsu and Blair, 2008*).

Dachs has not been shown to interact directly with Fat's ICD, and only three other proteins are known to affect Dachs accumulation in the subapical cell cortex, the casein kinase 1ε Discs overgrown (Dco), Approximated (App) and F-box-like 7 (Fbxl7). Dco may act through Fat itself: Dco binds and phosphorylates the Fat ICD, and loss of Dco function causes strong overgrowth and increases subapical Dachs, similar to loss of Fat (*Feng and Irvine, 2009*; Matakatsu, Blair and Fehon, Unpublished; *Pan et al., 2013*; *Rodrigues-Campos and Thompson, 2014*; *Sopko et al., 2009*).

App suggests a mechanism in which Fat inhibits the tethering of Dachs to protein complexes in the subapical domain (*Figure 1*). *app* mutants decrease subapical Dachs levels and reduce Dachs activity (*Matakatsu and Blair, 2008*). Thus, like *dachs* mutants, *app* mutants reverse the overgrowth and increased Yorkie target gene expression normally observed in *fat* mutants, and improve hair PCP. App is one of 20 Drosophila DHHC palmitoyltransferases, transmembrane proteins responsible

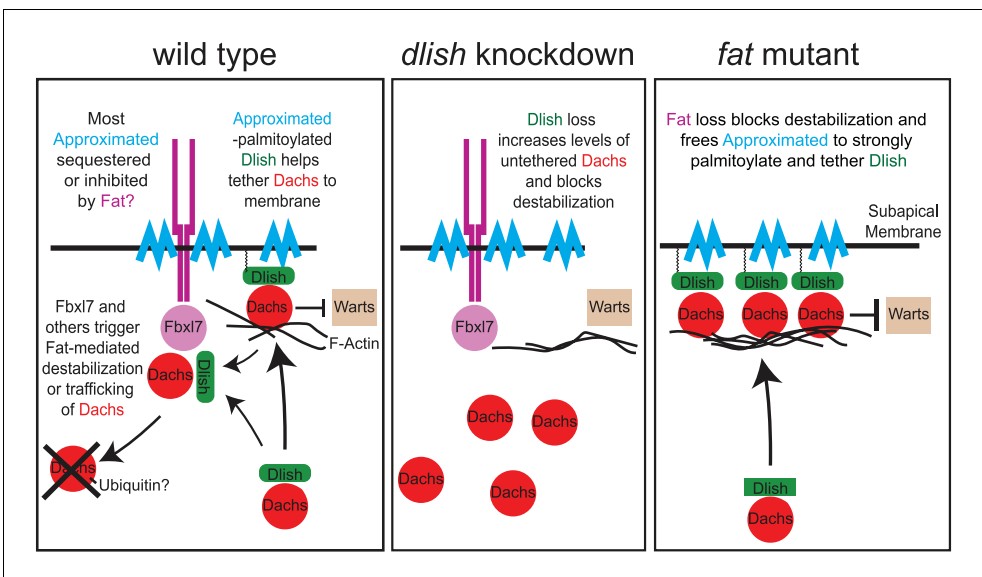

**Figure 1.** Models of Fat-mediated regulation of subapical Dachs by Fat-inhibited subapical tethering and Fat-stimulated destabilization. In wild type, Dlish binds Dachs, and when palmitoylated by Approximated helps tether Dachs to the subapical membrane where Dachs can inhibit Warts; Dachs contributes to tethering by binding the F-actin cytoskeleton. Fat binds Approximated and reduces its activity, reducing Dachs tethering. Fat also binds Fbxl7 and other ubiquitin ligases, destabilizing Dachs and trafficking it from the membrane. Dlish promotes the destabilization of Dachs, either indirectly by tethering it near Fat-bound ubiquitin ligases, or directly by helping Dachs complex with those ligases. In *dlish* mutants both the tethering and destabilization of Dachs are reduced; excess Dachs accumulates in the cytoplasm but no longer inhibits Warts. In *fat* mutants Approximated-mediated tethering increases and Fat-mediated destabilization of Dachs is lost, increasing subapical Dachs and the inhibition of Warts.

for adding palmitates to cytoplasmic proteins and thereby anchoring them to cell membranes (*Fukata et al., 2016*; *Linder and Deschenes, 2007*; *Nadolski and Linder, 2007*). App is also concentrated in the subapical cell membrane and can bind both Dachs and the Fat ICD (Matakatsu, Blair and Fehon, Unpublished). Thus, in the simplest model App palmitoylates or tethers Dachs, concentrating it in the cell cortex, and Fat works in part by sequestering or inhibiting App. However, Dachs is not detectably palmitoylated (*Matakatsu and Blair, 2008*; Matakatsu, Blair and Fehon, Unpublished).

Fbxl7 suggests a complementary mechanism based on ubiquitination. Fbxl7 binds Fat's ICD, and Fbxl7 localization and function are regulated by Fat; moreover, subapical Dachs increases in Fbxl7 mutants, causing moderate overgrowth (*Bosch et al., 2014*; *Rodrigues-Campos and Thompson, 2014*). Like other F-box proteins (*Skaar et al., 2013*), Fbxl7 could function in a Skp-Cullin1-F-box (SCF) ubiquitin E3 ligase complex, thereby targeting Dachs for proteasome-mediated degradation or trafficking away from the membrane (*Figure 1*). However, Fbxl7 has not been shown to bind Dachs or change its total levels in cells, and the two studies differ about its ability to stimulate Dachs ubiquitination. Nor does loss of Fbxl7 completely reproduce the strong overgrowth caused by loss of Fat or Dco. Nonetheless, the idea that Fat regulates a complex for the ubiquitination of Dachs is an attractive one (*Figure 1*), given Fat's ability to locally reduce the accumulation of Dachs. In fact, loss of Fat greatly increases the total levels of Dachs in imaginal disc cells (Matakatsu et al., Unpublished), a result we will confirm.

Below, we will describe the function of a novel Dachs binding protein, and show that its effects provide strong evidence for both the palmitoylation-dependent and degradation-dependent regulation of Dachs (*Figure 1*). Our structure-function analysis of Dachs found regions required for its normal subapical localization, and we used this information as the basis for a screen for novel Dachs binding partners. We found a direct binding partner for the Dachs C-terminus, the previously uncharacterized SH3 domain protein CG10933, which we have renamed <u>D</u>achs <u>li</u>gand with <u>SH</u>3s, or Dlish. We show that the activity and subapical concentration of Dlish are regulated by Fat, Dco and Dachs, and that Dlish in turn is required for the subapical concentration and full activity of Dachs in both wild type and *fat* mutant cells. Dlish localization also depends on App; furthermore Dlish binds to and is palmitoylated by App, and that palmitoylation can be suppressed by Fat. Loss of Dlish also increases the total levels of Dachs, likely by blocking Fat-mediated destabilization of Dachs. We propose that Dlish targets Dachs to subapical protein complexes in part via Fat-regulated, App-mediated palmitoylation. Dlish thereby concentrates Dachs where it can efficiently inhibit subapical Warts, and conversely links Dachs to the machinery for Fat-dependent destabilization.

## Results

### Hippo-specific subdomains of the Fat ICD affect the accumulation of subapical Dachs

Overexpression of a version of Fat that lacks its extracellular domain (FatΔECD) can rescue the overgrowth and heightened Yorkie target gene expression normally seen in *fat* mutant wings (*Matakatsu and Blair, 2006*). This rescue requires the presence of adjacent 'PH' and 'Hippo' domains in the ICD (*Matakatsu and Blair, 2012*) (see schematic in *Figure 2J*). We extended these studies by testing which domains reduce the increased subapical Dachs of *fat* mutants.

We found a strong correlation between constructs that rescued overgrowth and Hippo pathway activity and those that suppressed Dachs. *hh-gal4*-driven expression of a construct containing only the transmembrane, the PH and both the Hippo N and C domains (FtΔECD-PH+Hippo) reduced overgrowth, Yki targets and Dachs levels (*Figure 2A–C*), as did larger constructs containing these domains (*Figure 2D*) or a smaller construct containing only the Hippo domains (FtΔECD-Hippo) (*Figure 2—figure supplement 1*). Deletions lacking only one of the two Hippo domains could also weakly reduce Dachs levels (*Figure 2E,F*), in line with their weak effects on Hippo pathway markers (*Matakatsu and Blair, 2012*). Constructs lacking both the PH and Hippo domains did not detectably suppress subapical Dachs, even those that retained the N-terminal PCP-regulating region (FtΔECDΔ1-C; *Figure 2G*), or the region C-terminal to the Hippo domains (FtΔECDΔN-5 or ΔN-6; *Figure 2H,I*).

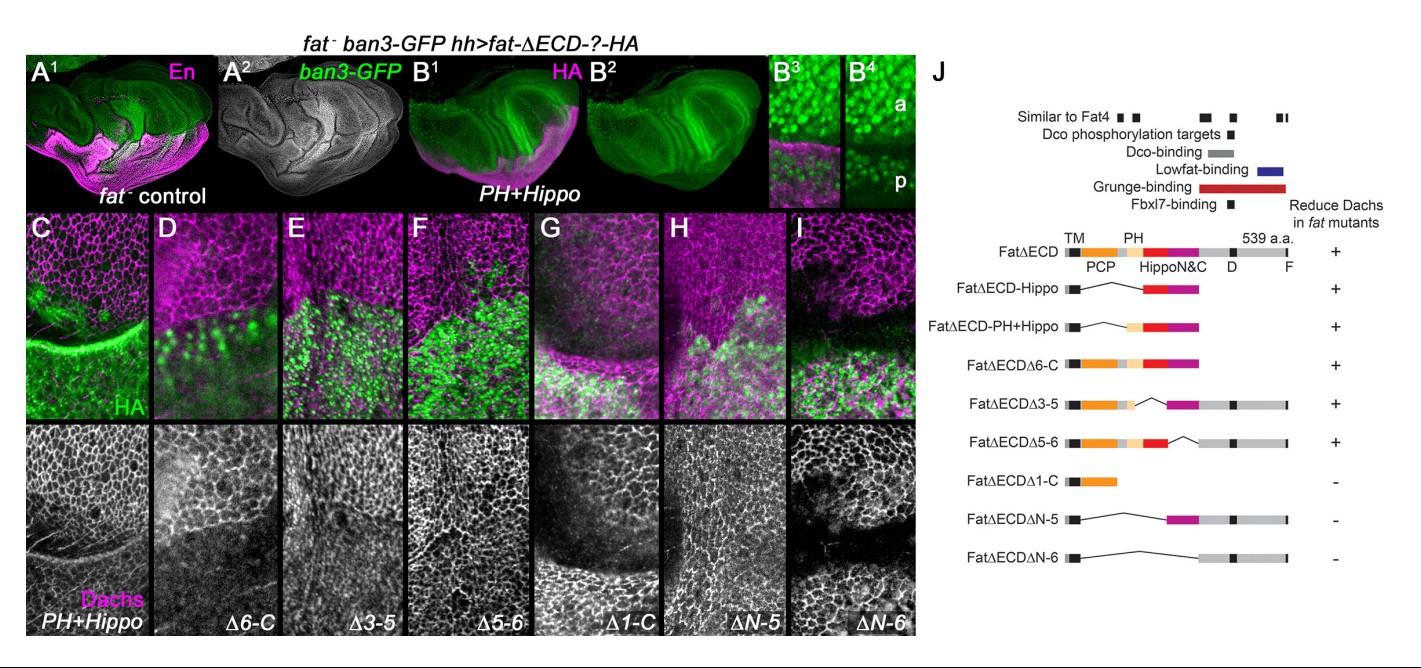

**Figure 2.** Regulation of Dachs levels by domains in the Fat ICD. (A[1],A[2]) Control *fat; hh-gal4 ban3-GFP* wing imaginal disc showing posterior with anti-Engrailed (En). Anterior and posterior are equally overgrown and the Yorkie activity reporter *ban3-GFP* is expressed at equally high levels in both regions. (B[1]–I) *fat* mutant wing imaginal discs expressing FatΔECD-HA constructs in the posterior with *hh-gal4*, identified using anti-HA. (B[1]–B[4]) Reduced overgrowth and *ban3-GFP* expression by posterior expression of FatΔECD-PH+Hippo. (C–I) Details of anterior-posterior boundary regions showing the posterior response of subapical Dachs to constructs (HA). (J) Summary of constructs and results.

The following figure supplement is available for figure 2:

**Figure supplement 1.** Effect of FatΔECD-Hippo on growth, *ban3-GFP* and Dachs.

This does not mean regions outside the Hippo domains lack function, especially at the lower expression levels of the endogenous gene. While point mutations in the Hippo domains cause overgrowth (*Bosch et al., 2014*; *Bossuyt et al., 2014*), *fat* mutants lacking the phosphorylation sites or the 'D' domain just C-terminal to Fat's Hippo domains (*Figure 2J*) also cause moderate overgrowth and increased Dachs levels, and deleting the C-terminal 'F' domain misorients Dachs polarization (*Bosch et al., 2014*; *Pan et al., 2013*; *Rodrigues-Campos and Thompson, 2014*). However, the D and F domains are not sufficient to visibly affect Dachs levels in the absence of the Hippo domains, even at the high levels of overexpression used here, suggesting a modulatory or cooperative role.

## Regions C-terminal to the Dachs myosin domain are critical for Dachs function and subcellular localization

To investigate the structures responsible for Dachs localization and activity, we tested whether expression of Dachs deletion constructs in *dachs* mutant flies mimicked the expression of full-length Dachs, which can rescue viability, increase the expression of Yorkie activity reporters such as *ban3-GFP* and concentrate at the subapical cell cortex (*Figure 3B,H*).

First, we found the portion of Dachs N-terminal to the myosin head was not needed for either function or subapical localization (D-ΔN; *Figure 3C,I*). This region contains a putative coiled-coil motif that is well-conserved in predicted Dachs homologs from other insects and some arthropods, but is missing from the shorter predicted homologs from most other taxa (*Figure 3—figure supplement 1*). D-ΔN does not, however, retain full Dachs activity, as expressing it in wild type flies slightly reduced growth and induced PCP defects, in contrast to the overgrowth caused by overexpressing full-length Dachs (*Figure 3—figure supplement 2C–F*). Antiserum specific to the Dachs N-terminus does not bind D-ΔN and so we used this to examine the effects of D-ΔN on endogenous Dachs.

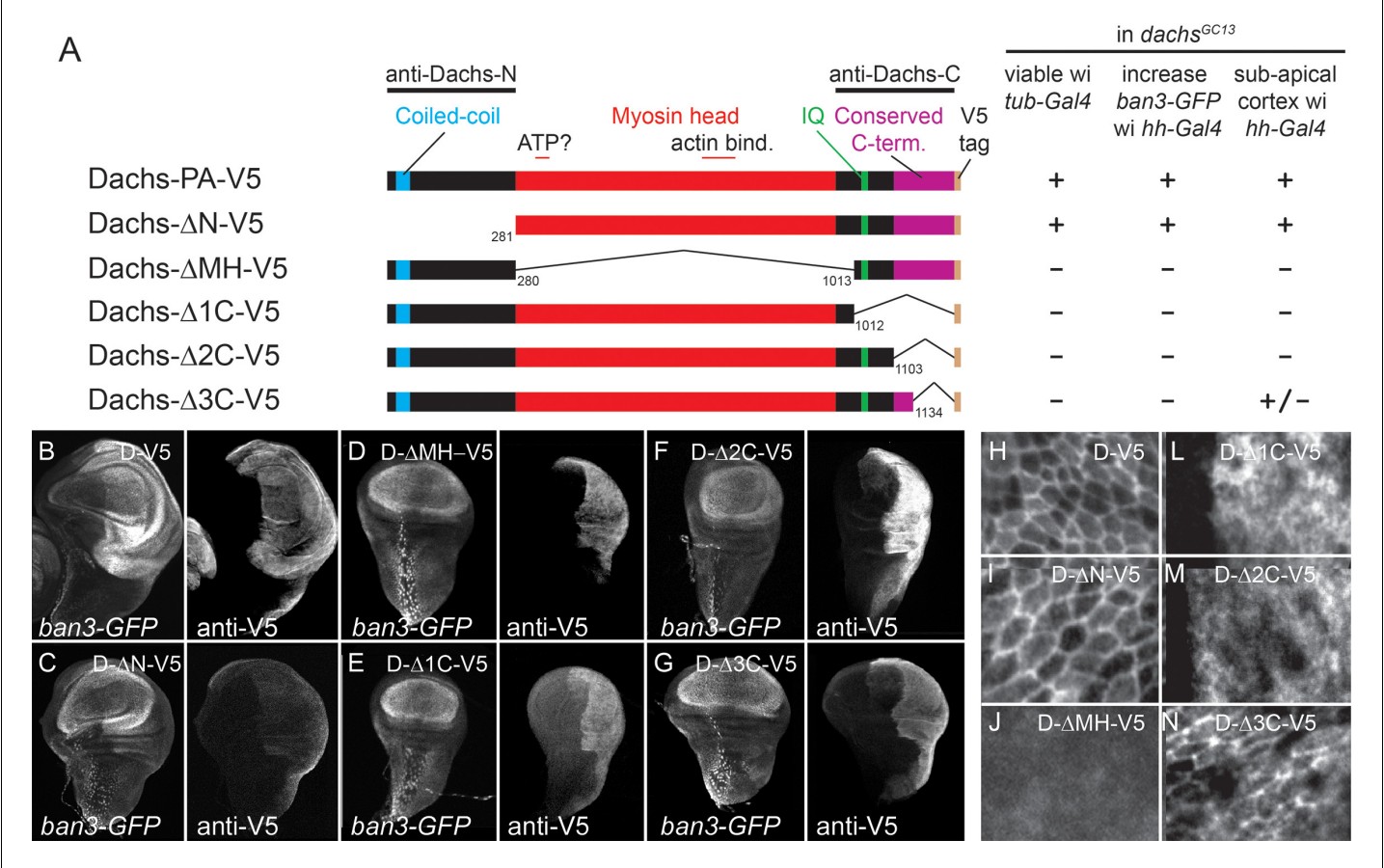

**Figure 3.** Structure-function analysis of Dachs. (**A**) Summary of Dachs-V5 deletion constructs and results of functional assays in *dachs* mutants. (**B–G**) Response of the Yki activity reporter *ban3-GFP* to the expression of Dachs-V5 deletion constructs in *dachs* mutant background; *ban3-GFP* is only increased by Dachs-V5 (**B**) and Dachs-ΔN-V5 (**C**). (**H–N**) Localization of Dachs-V5 constructs at subapical focal plane.

The following figure supplements are available for figure 3:

**Figure supplement 1.** Clustal omega alignment of N-termini and C-termini of *Drosophila melanogaster* Dachs-PA to predicted Dachs homologs.

**Figure supplement 2.** Effects of expressing truncated Dachs in wild type.

D-ΔN expression slightly decreased subapical and increased basolateral, cytoplasmic levels of endogenous Dachs (*Figure 3—figure supplement 2A*). We hypothesize that D-ΔN displaces endogenous Dachs from its normal binding partners in the apical cell cortex, replacing Dachs with the less effective D-ΔN form.

The Dachs myosin head domain binds F-Actin (*Cao et al., 2014*) and Warts (*Rauskolb et al., 2011*). Dachs lacking the myosin head (D-ΔMH) did not rescue *dachs* mutants (*Figure 3D*) and had no effect on wild type wings. D-ΔMH also localized to the cytoplasm, and did not concentrate in the sub-apical cell cortex (*Figure 3J*), consistent with the hypothesis that the head tethers Dachs to the cortical cytoskeleton.

C-terminal to the myosin domain is a single regulatory IQ domain that binds calmodulin (*Cao et al., 2014*) and several regions that lack Pfam and SMART motifs but are well-conserved in predicted homologs from all taxa (*Figure 3—figure supplement 1*). Removing the entire C terminus (D-Δ1C) impaired Dachs activity and localization, as did a smaller deletion that left the IQ domain intact (D-Δ2C) (*Figure 3E,F,L,M*). Removing only the C-terminal 98 amino acids (D-Δ3C) also impaired Dachs activity (*Figure 3G*); its concentration in the sub-apical cell cortex was slightly

improved compared with the larger C-terminal deletions, but it was still disorganized and cytoplasmic levels were abnormally high (*Figure 3N*). Overexpression of the C-terminal deletions did not affect the growth of wild type wings, and D-Δ1C did not affect the endogenous Dachs recognized by a C-terminal-specific anti-Dachs (*Figure 3—figure supplement 1B*).

## The SH3 protein Dlish directly binds to the Dachs C terminus

Given the importance of the C-terminal portions or Dachs, we used this region as bait in a yeast two-hybrid screen for potential binding partners. Of the 26 confirmed positives, 17 were to CG10933 (*Table 1*). CG10933 was also found to complex with Dachs in a mass-spec screen of the Hippo pathway interactome (*Kwon et al., 2013*), so we selected it for further study. We confirmed binding between Dachs and CG10933 by reciprocal co-immunoprecipitation (co-IP) in S2 cells (*Figure 4B*). The co-IP was specific for the C-terminus of Dachs (*Figure 4C*). Binding was direct, as we observed it in both yeast two-hybrid and by co-IP between GST-purified CG10933 and in vitro-translated Dachs constructs containing the C-terminus (*Figure 4E*). CG10933 binding to larger C-terminal Dachs constructs containing more of the myosin head was reduced, but still visible, possibly due to interference between the truncated head and the CG10933-binding region in the C-terminus.

CG10933 encodes a novel cytoplasmic protein containing three SH3 domains (*Figure 4A*). Predicted CG10933 homologs are found in many animal taxa, from Placozoa to Cephalochordata, that also contain a predicted Dachs homolog (*Figure 4—figure supplement 1*). Like Dachs, however, CG10933 lacks an obvious vertebrate ortholog. We have renamed CG10933 <u>D</u>achs <u>li</u>gand with <u>SH</u>3s, or Dlish.

The portion of Dachs that is deleted in D-Δ3C contains a proline-rich region that conforms to the consensus for both type I and type II SH3-binding domains (*Kaneko et al., 2008*), and that is conserved in most Dachs homologs (*Figure 3—figure supplement 1*). Co-IP depended on this binding domain (SH3Bd-3), but not two other candidate proline-rich sites, SH3Bd-1 and -2 (*Figure 4D*; *Figure 4—figure supplement 2A*). Conversely, in vitro translated Dlish-FLAG constructs lacking the second SH3 domain did not bind to GST-purified Dachs C-terminus (Dachs-Δ10), while Dlish constructs containing only the second SH3 domain bound strongly (*Figure 4F*); GST pulldown using GST-tagged Dlish fragments gave the same result (*Figure 4—figure supplement 2B*).

## Dlish complexes with the Fat ICD and its binding partners

In S2 cells, we have not detected binding between Dlish and the Dachs binding proteins Warts and Zyxin, the Warts-binding protein Ajuba, or the Hippo pathway components Hippo, Sav, Mats, or Yorkie. However, Dlish can co-IP with FatΔECD (*Figure 5A*). Known binding partners for the Fat ICD include Dco (*Feng and Irvine, 2009*; *Sopko et al., 2009*) App (Matakatsu, Blair and Fehon, Unpublished) and Fbxl7 (*Bosch et al., 2014*; *Rodrigues-Campos and Thompson, 2014*). Dco and App can co-IP Dlish (*Figure 5A*), as can Fbxl7 (see below). DsICD, a Ds construct which lacks both its

**Table 1.** Yeast two-hybrid screen with Dachs C-terminus

| 17 hits: |
| --- |
| *CG10933*- SH3 domain protein |
| **2 hits:** |
| *zld*- Zinc finger protein |
| **1 hit each:** |
| *AOX1*- Aldehyde oxidase 1 |
| *bc10*- Bladder cancer associated protein (BLCAP) homolog |
| *bent*- Myosin light chain kinase/Titin family (single Ig fragment) |
| *CG7220*- UBE2w-like E2 Ubiquitin-conjugating enzyme |
| *Maf1*- RNA polymerase III repressor |
| *Strn-Mlck*- Myosin light chain kinase/Titin family (single Ig fragment) |
| *Tep2*- Thioester-containing protein 2 |

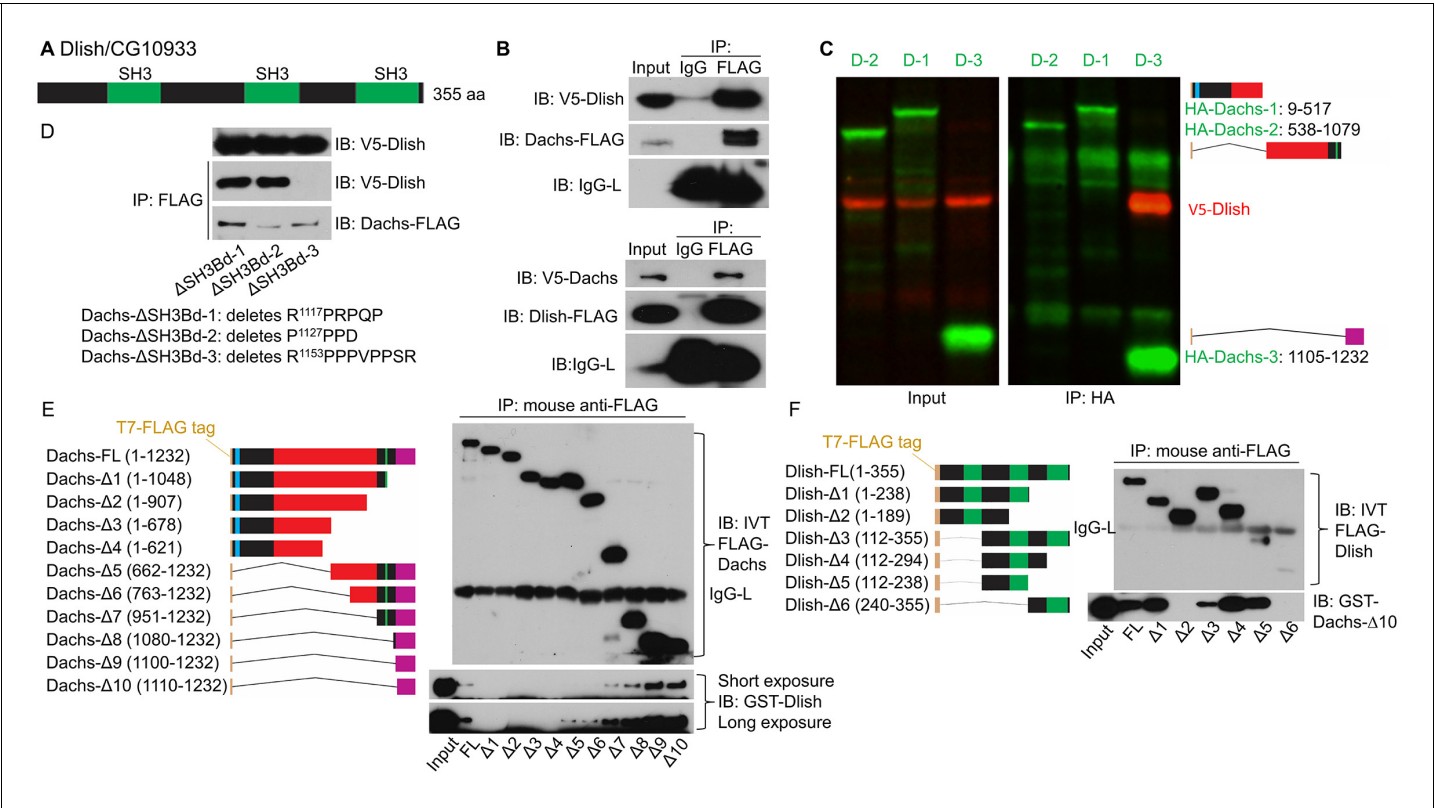

**Figure 4.** Binding between CG10933/Dlish and Dachs. (**A**) Schematic of Dlish domain structure. (**B**). Reciprocal co-IP between Dachs and Dlish from S2 cells. (**C**) Co-IP of V5-Dlish from CL8 cells only with the C-terminus of HA-Dachs. (**D**) Failure of Dlish to co-IP with Dachs from which the third of three candidate SH3 binding domains (SH3Bd) has been removed; the correspondence of SH3Bd-3 with the consensus for type I and type II binding domains in shown in *Figure 3—figure supplement 1*. (**E**) Co-IP of GST-purified Dlish only with in vitro-translated (IVT) FLAG-Dachs constructs containing the Dachs C-terminus. (**F**) Co-IP of GST-purified Dachs C-terminus (Dachs-Δ10) only with those in vitro translated FLAG-Dlish constructs containing the second of its three SH3 domains. See *Figure 4—figure supplement 2B* for equivalent GST pulldown.

The following figure supplements are available for figure 4:

**Figure supplement 1.** Clustal omega alignment of *Drosophila melanogaster* Dlish/CG10933 to predicted homologs.

**Figure supplement 2.** Additional data on binding between Dlish and Dachs.

extracellular and transmembrane domains, has been reported to co-IP Dachs (*Bosveld et al., 2012*), and can co-IP Dlish as well (*Figure 5A*).

While in vitro co-IPs should be treated with caution, these results raise the possibility that, in vivo, Dlish and its direct binding partner Dachs participate and function in the subapical protein complexes regulated by the ICDs of Fat and Ds. As summarized in *Figure 2J*, different domains of Fat's ICD have different binding and signaling activities, so we next examined Dlish binding to these domains. Fbxl7 and Dco bind to domains C-terminal to the Hippo domain (*Bosch et al., 2014*; *Feng and Irvine, 2009*; *Rodrigues-Campos and Thompson, 2014*; *Sopko et al., 2009*). We have found that Dlish can co-IP with the FatΔECDΔN-6 construct containing this region (*Figure 5B*). Dlish can also co-IP with FatΔECD-PH+Hippo, which includes the PH and Hippo domains most active in the regulation of growth and Dachs levels, but which had no previously known binding partners. Dlish does not co-IP with FatΔECDΔ4-C, which includes the PH but not the Hippo domains and which cannot regulate growth. We did not observe co-IP with the smaller FatΔECD-Hippo construct that lacks the PH domain. Although in vivo the PH region is not required for activity of the adjacent Hippo domains (e.g. *Figure 2—figure supplement 1*; *Matakatsu and Blair 2012*; *Pan et al. 2013*),

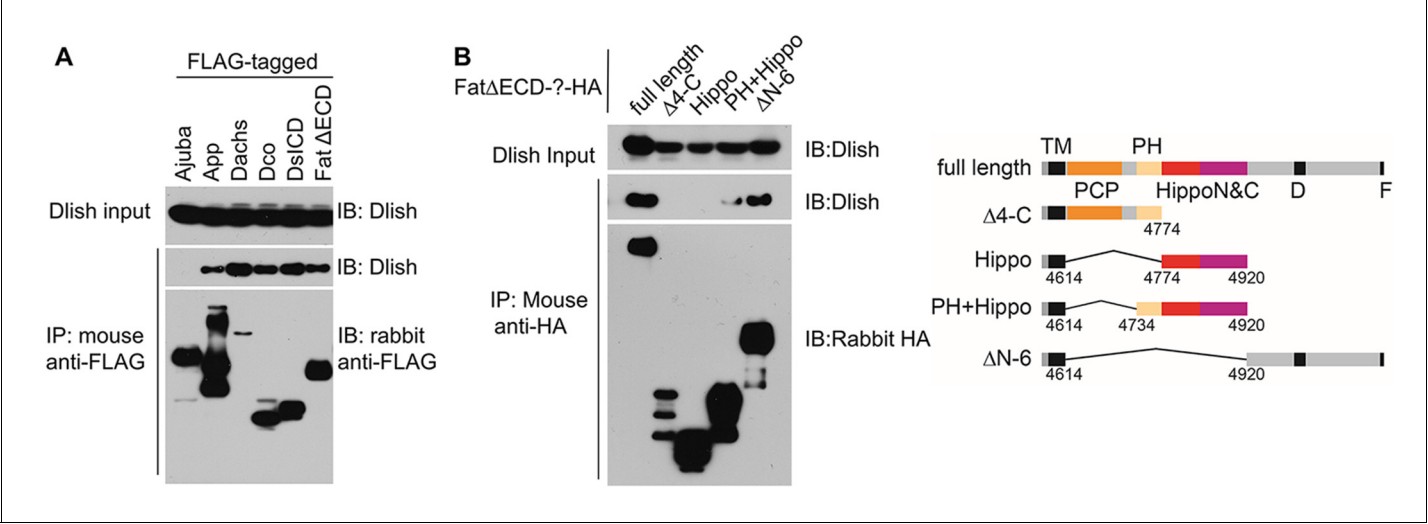

**Figure 5.** Dlish binding to Fat ICD and Fat pathway members. (A) Co-IP of Dlish in S2 cells with candidate Fat pathway proteins. Failure to bind Ajuba serves as a negative control. (B) Co-IP of Dlish from S2 cells with FatΔECD, FatΔECD-PH+Hippo and FatΔECDΔN-6, but not FatΔECDΔ4-C or FatΔECD-Hippo.

it does strengthen that activity (*Matakatsu and Blair 2012*), and in the different environment found in S2 cells may be more critical for proper folding or the formation of protein complexes.

## Fat and Dachs affect Dlish accumulation in the subapical cell cortex

We next examined how Dlish might be regulated by Fat and Dachs in vivo. In situ hybridization showed that Dlish is expressed ubiquitously in imaginal discs (*Figure 6A*). To analyze the subcellular distribution of Dlish we generated anti-Dlish. The reduced staining within regions of Gal4-driven *UAS-dlish-RNAi* allowed us to unambiguously localize endogenous Dlish: low levels of Dlish were found diffusely in the cytoplasm, but high levels were concentrated at the subapical cell cortex, overlapping the region of highest anti-Dachs staining (*Figure 6B,C*). Our best-stained preparations revealed a cell-by-cell polarization of Dlish, similar to that of Dachs (*Figure 6—figure supplement 1A*).

Not only does Dlish overlap Dachs in wild type wings, but Dlish shows a very Dachs-like response to the loss of Fat or Dco: Dlish levels in the subapical cortex increased dramatically in *fat* mutant clones, overlapping the similar accumulation of Dachs (*Figure 6D–F*), and we could no longer detect a consistent direction of Dlish polarization (*Figure 6—figure supplement 1B*). *dco*[3] mutant clones caused similar effects (*Figure 6—figure supplement 1C*). The Dlish increase was also observed in *fat* mutant discs, and was reduced by expression of FatΔECD-PH+Hippo (*Figure 6G*). The Dlish increase is not caused by the overgrowth and heightened Yorkie activity of *fat* or *dco* mutants, as *hh-gal4*-driven expression of an activated form of Yorkie (*UAS-Yki*[S168A]) caused strong, *fat* mutant-like overgrowth but had no effect on subapical Dlish (*Figure 6—figure supplement 1D*). The subapical Dlish accumulation is also regulated by Dachs, as subapical Dlish was increased by overexpression of Dachs (*Figure 6H*) and decreased in *dachs* or *fat dachs* mutant clones (*Figure 6I–J*).

## *dlish* is required for normal Dachs localization and *fat* mutant overgrowth

Not only does subapical Dlish depend on Dachs, but both Dachs localization and function depend on, and can be driven by, Dlish. Moderate reductions of Dlish led to adult phenotypes characteristic of reduced Dachs or increased Fat function. *dlish*[04], caused by insertion of a P element containing UAS-YFP.Rab26.Q250L into the first coding exon of the *dlish* gene (*Zhang et al., 2007*), is homozygous and hemizygous viable. Western blotting shows that *dlish*[04] still produces protein of approximately the normal size, but the placement of the P element insertion makes it likely that it is missing the six most N-terminal amino acids, which are highly conserved in homologs from other taxa

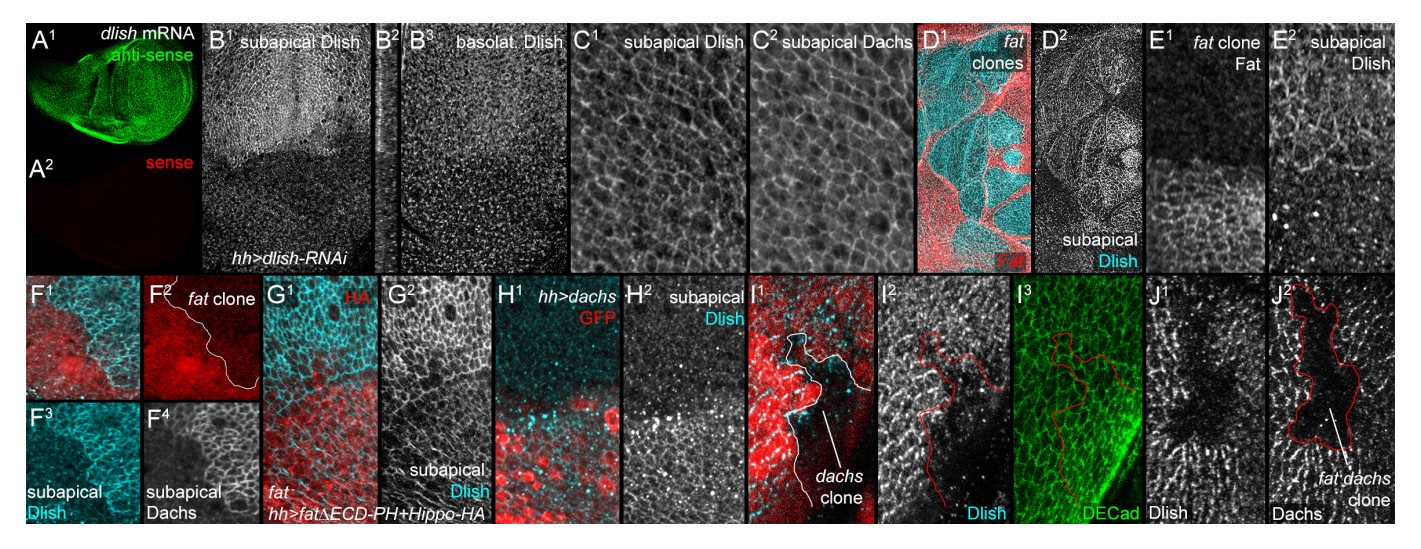

**Figure 6.** Localization of Dlish and its regulation by Fat and Dachs in wing imaginal discs. (A[1],A[2]) In situ hybridization to wing imaginal disc with anti-sense (A[1]) and sense (A[2]) probes. (B[1]–B[3]) *hh-gal4 UAS-dlish-RNAi* disc stained with anti-Dlish. Dlish remaining in anterior is concentrated subapically (B[1], left in cross-section in B[2]) and diffusely in the basolateral cytoplasm. (C[1],C[2]) Closeup showing overlap of Dlish and Dachs in the subapical cell cortex. (D[1]–F[4]) Homozygous *fat^fd* clones, shown by absence of anti-Fat stain (D[1],E[1]) or RFP (F[1],F[2]). Dlish is increased in the subapical cell cortex, coincident with Dachs (F[4]). (G[1],G[2]) Reduction of subapical Dlish in *fat* mutant disc by posterior, *hh-gal4*-driven expression of *UAS-fatΔECD-PH+Hippo-HA* (anti-HA). (H[1],H[2]) Increase in subapical Dlish by posterior, *hh-gal4*-driven expression of *UAS-dachs* (posterior identified with UAS-driven GFP, red) (I[1]–I[3]) *dachs^gc13* homozygous clone (outline) marked by absence of nuclear RFP (I[1]). Subapical Dlish is decreased, while DE-cadherin (I[3]) is unchanged. (J[1],J[2]) *fat^fd dachs^gc13* homozygous clone marked by absence of Dachs (outline in J[2]). Subapical Dlish (J[1]) is decreased.

The following figure supplement is available for figure 6:

**Figure supplement 1.** Additional data on the regulation of Dlish localization.

(*Figure 4—figure supplement 1*); subapical anti-Dlish staining is reduced in *dlish^04* homozygous cells (*Figure 7K*). As with reductions in *dachs* or overexpression of Fat, *dlish^04* homozygous and hemizygous wings had reduced spacing between the anterior and posterior crossveins (*Figure 7B*) and mild hair polarity defects, especially in the proximal wing (*Figure 7G*); legs had tarsal defects, and in the eye approximately 2% of the ommatidia had reversed polarity (*Figure 7—figure supplement 1D,E*). *dlish^4506*, a CRISPR-induced deletion, produced similar defects, as did two differently targeted *UAS-dlish-RNAi* constructs; in addition stronger RNAi knockdown (e.g. throughout the wing with *nubbin-gal4* or in the posterior of the wing with *hh-gal4* or *en-gal4*) reduced wing blade size, again similar to increased Fat or reduced Dachs function (*Figure 7C–E,H,I*; additional *dlish-RNAi* data in *Figure 7—figure supplement 1A–C*).

There was no obvious change in Fat or Ds levels or localization in *dlish^04* homozygous clones, or in regions of *UAS-dlish-RNAi*-mediated knockdown (*Figure 7—figure supplement 1F,G*). There was, however, a profound effect on anti-Dachs staining that was reminiscent of the Dachs mislocalization caused by the removal of its Dlish-binding C terminus. Dachs levels increased throughout the cytoplasm, and the concentration of Dachs at the subapical cell cortex was slightly to severely disrupted, depending on the strength of the Dlish knockdown (*Figure 7L–O*; additional examples in *Figure 7—figure supplement 1H,I*).

Dlish is also sufficient to change Dachs localization and activity. Overexpression of Dlish caused the overgrowth of adult wings (*Figure 7,J*), and in discs moderate increases in subapical Dachs and increased Yki activity (*Figure 7P,Q*). Dlish tagged at the C-terminus with FLAG was more effective than untagged Dlish despite both being expressed at similar levels, suggesting that the FLAG tag alters Dlish conformation or binding in a manner that increases its activity.

The effects of Dlish on subapical Dachs were even more striking in *fat* mutant cells. Instead of the strong increase normally observed in *fat* mutant tissue, subapical Dachs was reduced in the *dlish-*

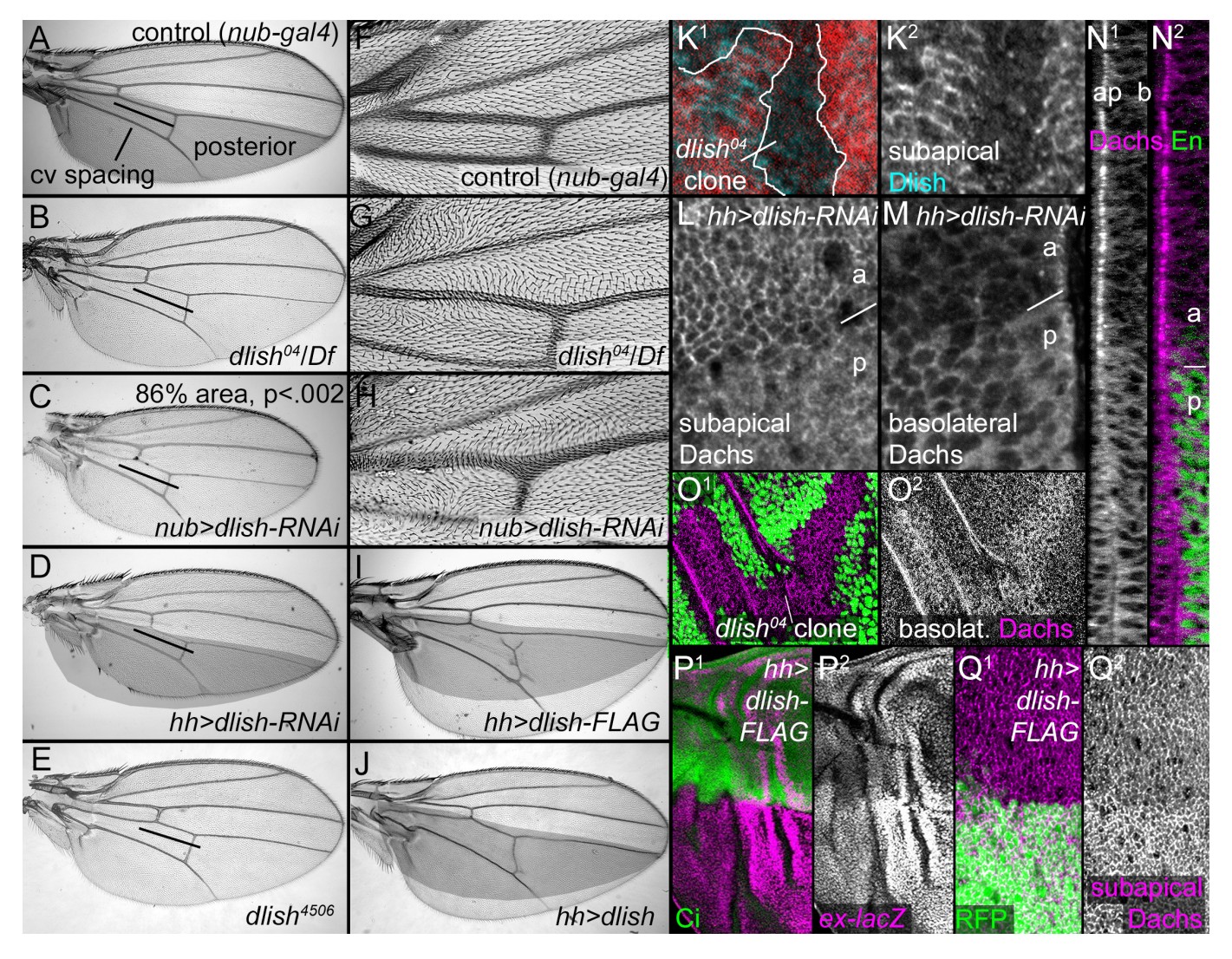

**Figure 7.** Dlish regulates Dachs function, localization and levels. (A–J) Adult wings. (A,F) Control, phenotypically wild type *nub-gal4* wing. The posterior compartment is marked in grey, and the bar marks the distance between anterior and posterior crossveins. (B,G). *dlish04*/*Df(2R)Exel7150* wing with reduced crossvein spacing (B) and proximal hair PCP defects (G). (C,H) *nub-gal4 UAS-dlish-RNAi* (VDRC) wing with reduced area (significant by Student's T and Whitney-Mann tests; see *Figure 7—source data 1* for data analysis) and PCP defects (H). (D) Posterior-specific, *hh-gal4*-driven *UAS-dlish-RNAi* (VDRC) reduces posterior size compared to grey control area from A. (E) *dlish4506* homozygous wing showing reduced crossvein spacing; hair PCP defects are similar to *dlish04* (detail not shown). (I,J) Posterior, *hh-gal4*-driven expression of *UAS-dlish-FLAG* (I) or *UAS-dlish* (J) increases posterior size compared to grey control area from A. (K1,K2) Reduction of subapical anti-Dlish staining in *dlish04* clone marked by absence of GFP (red). Because the GFP is in nuclei, the clone marker image is from the more basal, nuclear focal plane and slightly out of register with subapical Dlish. (L–N2) Posterior, *hh-gal4* driven expression of *UAS-dlish-RNAi* decreases cortical, subapical Dachs (L) and increases basolateral, cytoplasmic Dachs (M); N1 and N2 show the Dachs changes in the same cross-section Z series projection that shows Dlish in *Figure 6B2*, with Engrailed (En) marking the posterior (p) versus anterior (a). (O1,O2) *dlish04* homozygous clone, marked by absence of GFP, increases basolateral, cytoplasmic Dachs. (P1–Q2) Posterior, *hh-gal4*-driven expression of *UAS-dlish-FLAG* increases expression of the Yorkie activity reporter *ex-lacZ* (P1,P2) and increases subapical Dachs (Q1,Q2); anterior marked with Ci (P1) or posterior with GFP (Q1).

The following source data and figure supplement are available for figure 7:

**Source data 1.** Data and analysis for comparison of adult wing sizes between *nub-gal4* and *nub-gal4 UAS-dlish-RNAi* for *Figure 7C*.

**Figure supplement 1.** Additional data on *dlish* phenotypes.

*RNAi*-expressing region of *fat* mutant discs (*Figure 8A*). Moreover, *dlish-RNAi* also substantially reversed the overgrowth and increased Yki activity normally caused by loss of *fat* (*Figure 8A,B*), similar to the effects of *dachs* reduction in *fat* mutants. And in marked contrast to the severe disc overgrowth and 100% pupal lethality of *fat^{fd}* homozygotes, the majority of *dlish^{04} fat^{fd}* double homozygotes produced viable adults with mispatterned but nearly normally sized wings (*Figure 8C*).

In one view, the primary function of Dlish is to direct or tether Dachs to the subapical cell cortex; the cortical Dachs is then more active because it is better placed to bind and inhibit cortical Warts. Alternatively, Dlish could be directly mediating the regulation of Warts by Dachs, independent of its effects on Dachs localization. We favor the first alternative, as we have not found binding between Dlish and Warts or other Warts-binding proteins (see above). In addition, we found that tethering Dachs to the cell cortex by adding a CAAX prenylation site at its C-terminus can override the requirement for Dlish. Both *UAS-dachs* and *UAS-dachs-CAAX* induced similar overgrowth when expressed in the posterior of the wing with *hh-gal4* (*Figure 9A,B*). In a *dlish^{4506}* homozygous background, however, the overgrowth induced by Dachs overexpression was greatly reduced, while the

overgrowth induced by Dachs-CAAX was undiminished; there was a similar difference in their effects on the Hippo pathway marker *ban3-GFP* (*Figure 9C,D,G,H*). These different activities correlated with different protein localizations. In *dlish^{4506}* wing discs, overexpressed Dachs-CAAX was largely cortical (*Figure 9F*), while overexpressed Dachs was largely cytoplasmic (*Figure 9E*), contrasting with its cortical localization in wild type discs (*Figure 3H*).

## Dlish regulates total Dachs levels

Reduced Dlish activity not only affects the subcellular localization and activity of Dachs, but also the amount of Dachs in cells. *dlish^{04}* or *act-Gal4 UAS-dlish-RNAi* increased total Dachs levels, as assessed by western blotting protein from imaginal discs and normalized for cell number and size using α-Tubulin (*Figure 10A*; *dlish-RNAi* data in *Figure 10—figure supplement 1A*).

Loss of *fat* causes a similar increase in total Dachs levels in imaginal discs (Matakatsu, Blair and Fehon, Unpublished) (*Figure 10A*). Yet *dlish* knockdown increases cytoplasmic Dachs, whereas in *fat* mutants the increase is subapical. To explain this, we hypothesize that as Dlish targets or stabilizes Dachs in the subapical cell cortex, it places Dachs in or near complexes where it can also be destabilized by Fat (*Figure 1*). When Fat is lost destabilization no longer occurs, so the total and subapical levels of Dachs increase. When *dlish* is reduced, the Fat-dependent destabilization of Dachs cannot occur but subapical targeting is also lost, and so Dachs accumulates in the cytoplasm.

If the loss of Dlish increases Dachs levels by blocking Fat-mediated destabilization, then it will reduce the sensitivity of total Dachs levels to the presence or absence of *fat*; the effects of removing both Fat and Dlish will not be additive. Our data agree with this prediction. In

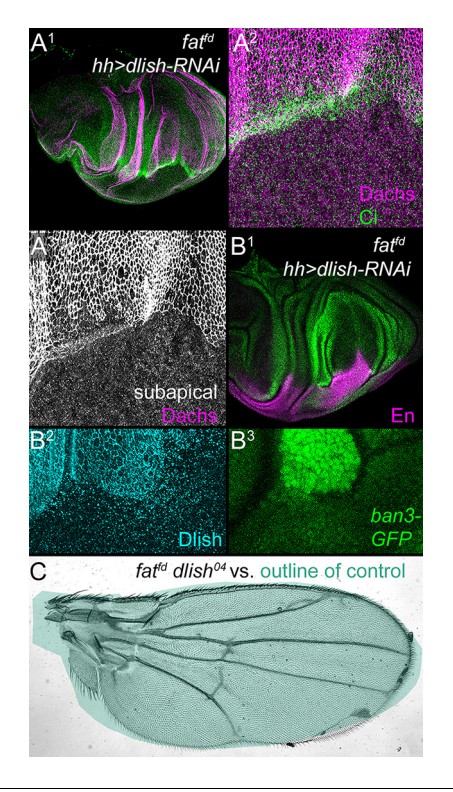

**Figure 8.** Reduced Dlish function suppresses the overgrowth and Dachs upregulation of *fat* mutants. (A^1–B^3) *fat^{fd}* discs with posterior, *hh-gal4*-driven expression of *UAS-dlish-RNAi*. Anterior marked with Ci (A^1,A^2), or posterior with En (B^1) or Dlish loss (B^2). Overgrowth is reduced in the posterior (A^1,B^1, compare with *Figure 2A*), as is subapical Dachs (A^2,A^3) and the Yorkie activity reporter *ban3-GFP* (B^1,B^3). (C) *fat^{fd} dlish^{04}* double mutant wing is abnormally patterned but nearly-normally sized when compared with the green outline of a control wing from *Figure 7A*. Unlike the 100% lethality of *fat^{fd}*, 50–60% of *fat^{fd} dlish^{04}* survive to adulthood (44/230 from self-crossed *dlish^{04} fat^{fd}* / *CyO-TM6,Tb* vs. 77/230 expected from self-crossed *non-lethal* / *CyO-TM6,Tb*).

trials using the hypomorphic *dlish*[04] background, simultaneous loss of *fat* caused no or only a weak additional increase in total Dachs levels compared those caused by *dlish*[04] alone; the Dachs increase in *dlish*[04] *fat*[fd] double homozygotes was similar to that seen in *fat*[fd] mutants, and significantly weaker than the increase expected if the *fat* and *dlish* effects were additive (*Figure 10A*). Similar results were obtained in a smaller number of trials combining *fat* loss with *UAS-dlish-RNAi* driven by *tubulin promoter (tub)-gal4* or *actin promoter (act)-gal4* (*Figure 10—figure supplement 1A*).

While screening for candidates that might also regulate Dachs levels, we found that Dlish can co-IP in vitro with the Fat-regulated E3 ubiquitin ligase Fbxl7, the related F-box protein Slimb and their binding partner Cullin1 (*Figure 10B*). Loss of Fbxl7 increases the levels of subapical Dachs in imaginal discs (*Bosch et al., 2014*; *Rodrigues-Campos and Thompson, 2014*), and we found a similar effect on subapical Dlish (*Figure 10C*). Homozygous *slimb*[1] clones also slightly increased subapical Dachs and Dlish (*Figure 10D,E*), as well as the known Slimb target Expanded (*Ribeiro et al., 2014*; *Zhang et al., 2015*). Dlish and Fat-binding ubiquitin ligases therefore help regulate the levels and localization of the Dlish-Dachs complex, providing one route for Fat's regulation of pathway protein levels (see Discussion).

## Approximated binds and regulates Dlish

We next addressed how changes in Dlish might help direct or tether Dachs to the subapical cell cortex. As noted above we found that Dlish can co-IP the transmembrane protein App (*Figures 5A*, *11A*). Dlish-App binding was confirmed in a GST pulldown of App produced by S2 cells, and requires the Dlish N-terminus containing the first SH3 domain- (*Figure 11B*). Since the second SH3 domain of Dlish binds Dachs (*Figure 4F*), Dlish could bridge App and Dachs using different domains.

This was particularly intriguing given the subapical localization of App, and the similarity between the effects of *dlish* and *app* knockdown on Dachs localization, levels and activity in wild type and *fat* mutant tissue (see Introduction) (*Matakatsu and Blair, 2008*; Matakatsu et al., Unpublished). This raised the possibility that App acted through Dlish and thus the Dlish-Dachs complex. In support of this hypothesis, we found that clonal loss of *app* in *fat* mutant wing discs decreased subapical and increased cytoplasmic Dlish; loss of *app* in wild type discs also increased cytoplasmic Dlish (*Figure 11C–E*). Dlish and App also showed a suggestive genetic interaction: while overexpression of App slightly reduced wing growth in wild type wings, it significantly strengthened the overgrowth caused by overexpression of a weakly active untagged Dlish (*Figure 11F–J*). Overexpression of *app* has a similar effect on the overgrowth of *dachs*-overexpressing wing discs (*Matakatsu and Blair, 2008*). Thus App increases the effectiveness of the Dlish-Dachs complex.

As a subapical transmembrane protein, App could be acting as a physical tether for Dlish and Dachs; however, App is also a DHHC palmitoyltransferase, and so might palmitoylate and help anchor the Dachs-Dlish complex near the membrane. Dachs is not detectably palmitoylated in vitro or in vivo (*Matakatsu and Blair, 2008*; Matakatsu, Blair and Fehon, Unpublished). We therefore directly examined the palmitoylation of Dlish using the Acyl Biotinyl Exchange (ABE) method, which uses N-ethylmaleimide to block free cysteines, hydroxylamine (HAM) to break palmitoyl-thioester bonds, and sulfhydryl-reactive biotin to label those bonds (*Drisdel et al., 2006*; *Drisdel and Green, 2004*). We detected HAM-dependent biotinylation of Dlish in S2 cells and, as a positive control, App itself (Matakatsu, Blair and Fehon, Unpublished), but not of the negative control GFP (*Figure 12A*). Moreover, HAM-dependent biotinylation of Dlish was increased by co-expression with App (13 of 14 repeats; examples are shown in *Figure 12A,C*, and averages and significance are shown in *Figure 12D*). This strongly argues that a substantial fraction of the biotinylation represents App-sensitive palmitoylation.

App can also bind to the ICD of Fat (Matakatsu, Blair and Fehon, Unpublished). We have confirmed binding in S2 cells, and found that App can co-IP not only with the domains C-terminal to the Hippo domains (FatΔECDΔN-6), but also the domains most strongly active in Dachs and Hippo pathway regulation (FatΔECD-PH+Hippo) (*Figure 12B*). Like Dlish, App failed to bind to FatΔECDΔ4-C and FatΔECD-Hippo, which have, respectively, no or reduced Hippo activity in vivo (*Matakatsu and Blair, 2012*). These results raise the possibility that Fat binding might regulate the activity of App on substrates like Dlish.

We therefore tested whether the addition of FatΔECD could alter App-stimulated Dlish palmitoylation in S2 cells, and found a marked decrease compared with the addition of App alone (6 of 6

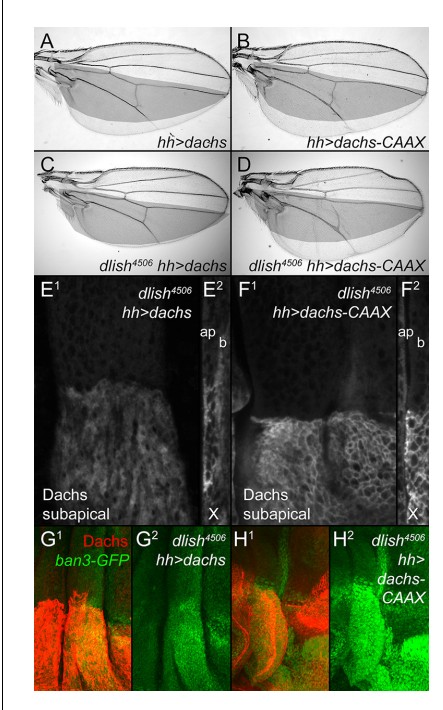

**Figure 9.** Dlish is required for the function of overexpressed Dachs but not Dachs-CAAX. (**A**,**B**) Similar overgrowth induced by posterior, *hh-gal4*-driven expression of *UAS-dachs-V5* (**A**) and *UAS-dachs-CAAX* (**B**). (**C**,**D**) Overexpression in *dlish⁴⁵⁰⁶* homozygote largely blocks overgrowth induced by *UAS-dachs-V5* (**C**) but not *UAS-dachs-CAAX* (**D**). (**E**,**F**) Comparison of anti-Dachs staining in *dlish⁴⁵⁰⁶* wing discs after posterior overexpression of *UAS-dachs-V5* (**E**) or *UAS-dachs-CAAX* (**F**), showing subapical focus (**E**¹,**F**¹) or z-series cross-section (**E**²,**F**², apical left). Cortical localization of Dachs-V5 is lost (compare with *Figure 3H*), but of Dachs-CAAX is retained. Discs were fixed and stained in parallel and imaged with identical settings. (**G**,**H**) Comparison showing weaker posterior upregulation of the Hippo activity marker *ban3-GFP* in *dlish⁴⁵⁰⁶* wing discs by posterior overexpression of *UAS-dachs-V5* (**G**) than by *UAS-dachs-CAAX* (**H**), imaged using the same confocal settings for GFP.

repeats; example in *Figure 12C*; averages and significance in *Figure 12D* and *Figure 12—source data 1*). This result is consistent with the model that Fat reduces subapical Dachs by interfering with or modifying the interaction between App and Dlish, reducing App's ability to palmitoylate Dish and tether the Dlish-Dachs complex.

## Discussion

The unconventional myosin Dachs is an important effector Fat/Ds-regulated Hippo signaling, as its heightened subapical levels in *fat* mutants inhibit and destabilize Warts, freeing Yorkie to increase the expression of growth-promoting genes (*Cho et al., 2006*; *Rauskolb et al., 2011*; *Vrabioiu and Struhl, 2015*). We used a structure-function analysis of Dachs as a springboard to search for new binding partners that are critical for Dachs localization and function, and have found Dlish (CG10933), a novel SH3 domain protein. Dlish binds directly to the Dachs C-terminus, and loss of Dlish disrupts Dachs localization, levels and function: subapical accumulation of Dachs is reduced and cytoplasmic and total levels increase, both in wild type and *fat* mutant tissue, while activity is lost. Importantly, Dlish is regulated by Fat, as loss of Fat greatly increases Dlish levels in the subapical cell cortex and, like Dachs, Dlish is needed for much of the *fat* mutant overgrowth.

Dlish also binds the ICD of Fat and other Fat-binding proteins, including two that likely mediate part of its function: the palmitoyltransferase App and the F-box protein Fbxl7. Thus Dlish provides a new biochemical link from the Fat ICD to Dachs regulation. Our evidence indicates that Dlish plays two different and opposing roles (*Figure 1*). First, it helps tether Dachs in the subapical cell membrane, in part via Fat-regulated, App-dependent palmitoylation, so that Dachs can more efficiently inhibit Warts. Second, it links Dachs to Fat-organized machinery for Dachs destabilization, including Fbxl7, and thus helps reduce Dachs levels.

### Dlish, App and subapical tethering

Dlish and Dachs cooperate to target or tether the Dlish-Dachs complex, as each is necessary, and to a weaker extent sufficient, for the subapical concentration of the other. The Dachs contribution is likely through tethering the complex to the cortical cytoskeleton, as we found that loss of the F-Actin-binding myosin head blocks the subapical localization of Dachs. This would agree with recent biochemical analyses that suggest that Dachs has no motor function, acting rather as an F-Actin-binding scaffolding protein (*Cao et al., 2014*).

The Dlish contribution, on the other hand, depends at least in part on its ability to bind the transmembrane DHHC palmitoyltransferase App. Loss of App (*Matakatsu and Blair, 2008*; Matakatsu, Blair and Fehon, Unpublished) and Dlish have very similar effects on Dachs localization and activity.

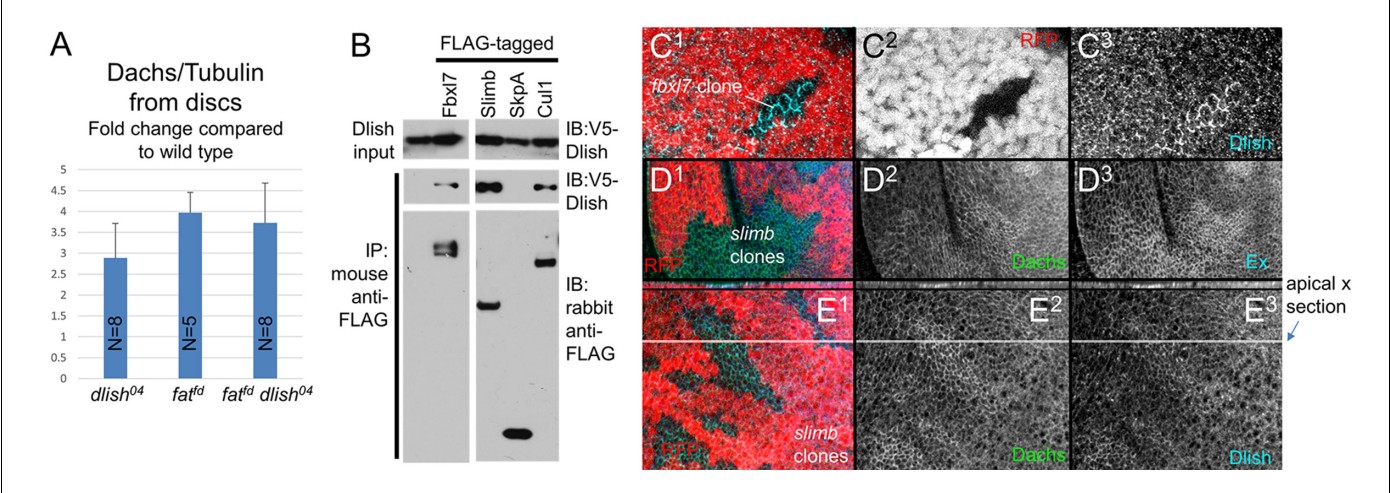

**Figure 10.** Regulation of Dachs and Dlish levels. (A) Dachs levels in extracts from wing imaginal discs, normalized to α-Tubulin levels and compared to wild type from the same trial. All Dachs changes were significantly different from wild type, but not from each other. Dachs levels in *dlish fat* double mutants are significantly lower than the predicted additive effects of *dachs* and *fat* (two-tailed p<0.05 using the Wilcoxon Rank Sum test, p=0.000036 using a single sample T test; see *Figure 10—source data 1* for data and analysis). (B) Co-IP of Dlish in S2 cells with F-box proteins Fbxl7 and Slimb and with Cullin 1, but not SkpA, from a single exposure of a single blot. See *Figure 10—figure supplement 1* for more examples. (C$^1$–E$^3$) Subapical proteins in mutant clones, marked by absence of RFP. RFP is from the nuclear, more basal focal plane, slightly out of register with subapical clone boundaries. (C$^1$–C$^3$) Increased subapical Dlish in *fbxl7* clone. (D$^1$–D$^3$) Increased subapical Dachs and Expanded (Ex) in *slimb$^1$* clones. (E$^1$–E$^3$) Increased subapical Dachs and Dlish in *slimb$^1$* clones; subapical cross sections from marked line are shown above each frame.

The following source data and figure supplement are available for figure 10:

**Source data 1.** Data and analysis of changes in Dachs in imaginal disc extracts for *Figure 10A*.

**Figure supplement 1.** Additional data on Dachs levels and Dlish binding.

We found that loss of App disrupts the subapical accumulation of Dlish in vivo, and that App can stimulate palmitoylation of Dlish in vitro. Thus, palmitoylation of Dlish likely stimulates membrane association of both Dlish and its binding partner Dachs.

App also has additional effects on Fat pathway activity. First, App has palmitoyltransferase-independent activity and can co-IP Dachs in vitro (Matakatsu, Blair and Fehon, Unpublished). Thus, while palmitoylation of Dlish may mediate some of App's activity, subapical App may simultaneously help localize the Dlish-Dachs complex by physical tethering. And while both palmitoylation and tethering of the Dlish-Dachs complex is likely critical for the *fat* mutant phenotype, App also has a function that depends on the presence of Fat, as App can bind, palmitoylate and inhibit the activity of Fat's ICD (Matakatsu, Blair and Fehon, Unpublished).

An important question is whether the absence of Fat regulates the App-dependent tethering of the Dlish-Dachs complex. The Fat ICD can complex with both Dlish (this study) and App (Matakatsu, Blair and Fehon, Unpublished). We found that Dlish and App can bind not only the C-terminal region of the Fat ICD where Fat is palmitoylated, but also the PH and Hippo domains which we showed played the strongest role in Dachs regulation. An attractive mechanism is that Fat inhibits the interaction between App and Dlish, reducing App's ability to palmitoylate and tether the Dlish-Dachs complex. In the absence of Fat, App and Dlish are freed to tether Dachs, and Dachs now inhibits and destabilizes Warts, causing overgrowth (*Figure 1*). In support of this model, we found that overexpression of Fat's ICD in vitro can reduce App-stimulated palmitoylation of Dlish.

## Dlish and the regulation of Dachs levels

Our evidence further indicates that Dlish targets Dachs for Fat-dependent destabilization. Loss of Fat increases not only subapical Dachs, but also total Dachs levels (Matakatsu, Blair and Fehon, Unpublished), a result we have confirmed. In the presence of Dlish the increased Dachs remains

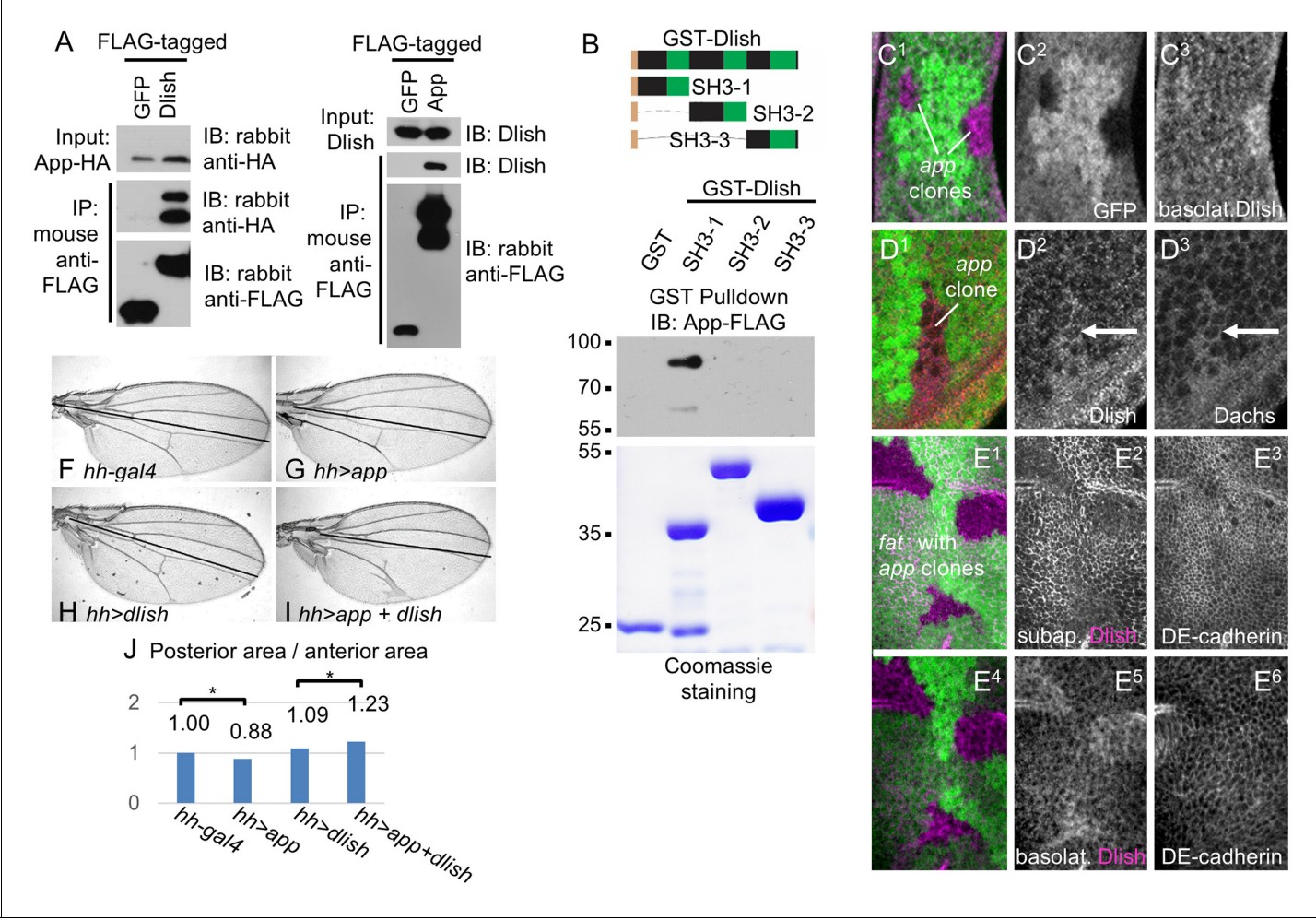

**Figure 11.** Dlish binds to and is regulated by App. (**A**) Reciprocal co-IP of App-HA by Dlish-FLAG and Dlish by App-FLAG. (**B**) Pulldown of S2 cell-generated App-FLAG by a GST-Dlish construct containing its first SH3 domain. (**C¹–D³**) Increased cytoplasmic Dlish and Dachs in homozygous *app* clones marked by absence of GFP. (**E¹–E⁶**) *app* clones marked by absence of GFP in a *fat* mutant wing disc. Subapical Dlish levels decrease (**E²**) and basolateral, cytoplasmic levels increase (**E⁵**), while control DE-cadherin is unchanged (**E³,E⁶**). (**F–J**) Posterior, *hh-gal4*-driven expression of *UAS-app* decreases the wing area in a wild type wing, but increases the overgrowth induced by *UAS-dlish* (untagged). Brackets show p<0.01 by a single-tailed Student's T and Whitney-Mann tests; data and analysis shown in *Figure 11—source data 1*.

The following source data is available for figure 11:

**Source data 1.** Data and analysis of wing size change after *hh-gal4*-driven expression of *UAS-app, UAS-dlish* (untagged), or both, for *Figure 11J*.

subapical. Loss of Dlish also increases the total levels of Dachs, but now that increase is cytoplasmic, and much less effective at inhibiting Warts. These Dachs increases are unlikely to have independent causes, as they are not additive; total Dachs levels are similar after loss of Fat, Dlish or both.

We propose that in wild type cells there is a flux of the Dachs-Dlish complex from the cytoplasm to the subapical cell cortex, where a Fat-dependent complex destabilizes Dachs (*Figure 1*). Normally the tethering effects of Dlish predominate over the Fat-dependent destabilization, and moderate levels of subapical Dachs are maintained. Destabilization is lost without Fat; this combines with Dlish-mediated tethering to increase subapical Dachs. Without Dlish the subapical tethering of Dachs is disrupted, access to Fat-dependent destabilization is lost and the now cytoplasmic Dachs increases. The model thus explains why the excess, largely cytoplasmic Dachs caused by reduced Dlish function is not greatly influenced by the presence or absence of Fat.

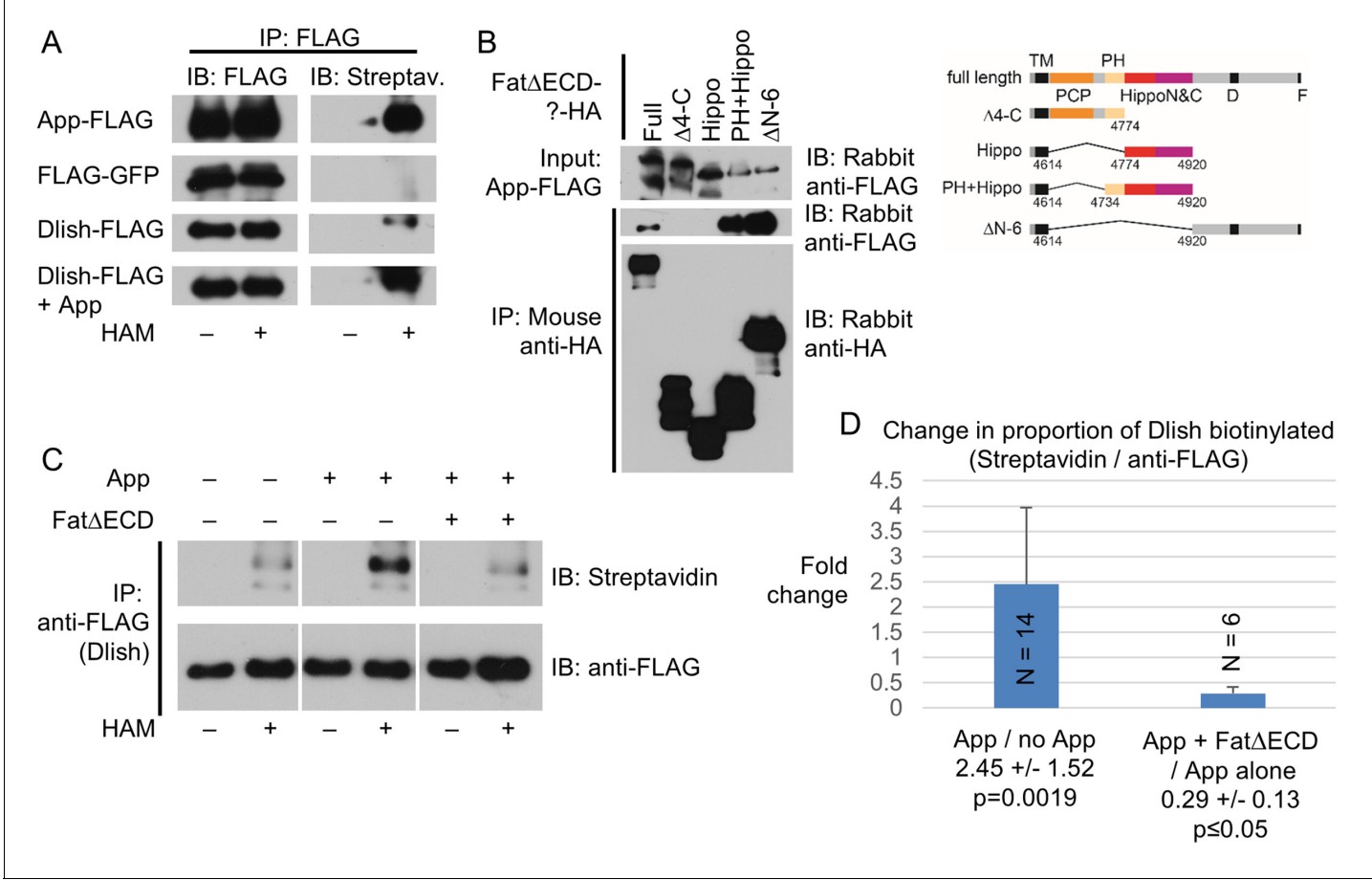

**Figure 12.** Dlish palmitoylation is regulated by App and the Fat ICD. (**A**) ABE assay for Dlish palmitoylation in S2 cells. All anti-FLAG or streptavidin stained lanes are from the same exposure of a single blot. HAM increases streptavidin labeling of Dlish and positive control App, but not negative control GFP. Presence of additional App in Dlish-expressing cells increases labeling. (**B**) Co-IP of App-FLAG with HA-tagged domains of the Fat ICD co-expressed in S2 cells. (**C**) ABE assay for Dlish palmitoylation in S2 cells. All anti-FLAG or streptavidin-stained lanes are from the same exposure of a single blot. Presence of additional App increases labeling, but the presence of App and FatΔECD reduces labeling compared with App alone. (**D**) Quantification of the changes in the proportion of Dlish that is biotinylated, expressed as the average fold-change between two different conditions in individual trials. The presence of additional App significantly increases Dlish labeling compared with no added App; the presence of App and FatΔECD significantly decreases Dlish labeling compared with App alone. p values are from two-tailed Wilcoxon Rank Sum tests to a hypothetical median of 1. For data and Wilcoxon and T test analyses see **Figure 12—source data 1**.

The following source data is available for figure 12:

**Source data 1.** Data and analysis of ABE assays for **Figure 12D**.

In addition to any effects caused by changing the subcellular localization of Dachs, Dlish may also provide a direct link to the machinery for protein ubiquitination, as Dlish can co-IP with the E3 ubiquitin ligase Fbxl7, as well as the related F-box ubiquitin ligase Slimb. Fbxl7 is particularly intriguing, as it binds to and is regulated by Fat's ICD, and reduces subapical Dachs, perhaps via ubiquitination (*Bosch et al., 2014*; *Rodrigues-Campos and Thompson, 2014*). Slimb can bind and ubiquitinate Expanded (*Ribeiro et al., 2014*; *Zhang et al., 2015*), a subapical regulator of Hippo signaling with links to Fat and Dachs function (*Irvine and Harvey, 2015*). But while we found that loss of Fbxl7 or Slimb increases subapical Dachs and Dlish, these effects are weak (see also *Bosch et al., 2014*; Matakatsu, Blair and Fehon, Unpublished), and the large increase in total Dachs levels caused by loss of Dlish or Fat must involve additional partners.

## Fat, Dachs and Dlish in other taxa

Mutations in Fat's closest mammalian homolog Fat4 (FatJ) and its Ds-like ligands strongly disrupt PCP-like processes, and have in humans been associated with the multisystem defects of Hennekam and Van Maldergem syndromes (*Alders et al., 2014*; *Mao et al., 2011*; *Saburi et al., 2008*; *Zakaria et al., 2014*). There has been some debate, however, about whether the mammalian proteins retain direct regulation of Hippo activity (*Bagherie-Lachidan et al., 2015*; *Bossuyt et al., 2014*; *Pan et al., 2013*; *Sadeqzadeh et al., 2014*; *Kuta et al., 2016*). Nonetheless, Fat4 has been linked to Hippo changes in both normal development and tumors (*Qi et al., 2009*; *Van Hateren et al., 2011*), mutations in Fat4 or Dachsous1 change the balance of precursors and mature neurons in the developing neuroepithelium of both humans and mice, and the mouse defect can be reversed by knockdown the Yki homolog Yap (*Cappello et al., 2013*). But the mechanisms underlying these effects are unknown, and Fat4 cannot regulate Hippo signaling in Drosophila (*Bossuyt et al., 2014*).

It is therefore important to note that while homologs of Dachs and Dlish are found throughout the animal kingdom, they are apparently absent from vertebrates. This suggests that the Dachs-Dlish branch of the Fat-Ds pathway, with its powerful effect on Warts activity, is also lacking. Nonetheless, it has been suggested that Drosophila Fat and Ds can affect Hippo pathway activity in a Dachs-independent manner (*Degoutin et al., 2013*; *Gaspar et al., 2015*). It is also clear that Drosophila Fat has Dachs-independent effects on PCP; indeed the N-terminal 'PCP' domain of the Fat ICD that did not affect Dachs in this study is sufficient to improve the PCP defects of *fat* mutants (*Matakatsu and Blair, 2012*). These or alternative pathways may still be present in mammals.

# Materials and methods

## Fly strains

*ds::GFP* (*Merkel et al., 2014*) was kindly provided by S. Eaton
*FRT$^{82B}$ fbxl7$^{C616Y}$* (*Bosch et al., 2014*) was kindly provided by I. Hariharan
*FRT$^{82B}$ slmb$^1$* (*Ribeiro et al., 2014*) was kindly provided by N. Tapon.

### From or modified from the Bloomington Drosophila Stock Center
*UAS-GFP; hh-gal4/TM6B,Tb*
*act-gal4/TM6B,Tb*
*tub-gal4/TM3,Sb Ser*
*nub-gal4*
*ap-gal4 UAS-GFP/CyO*
*ptc-gal4 UAS-GFP*
*y w hsFLP; FRT$^{2A}$ ubi-GFP*
*y w hs-FLP; ubi-RFP FRT$^{40A}$/CyO*
*y w hs-FLP; ubi-RFP FRT$^{42D}$/CyO*
*y w hs-FLP; ubi-RFP FRT$^{82B}$*
*CG10933$^{04}$*
*Df(2R)Exel7150/CyO*
*UAS-dlish-RNAi TRiP.HMS01708*
*y$^1$ w* P[nos-phiC31int.NLS]X; PBac[y+-attP-9A]VK00027*

### From the VDRC
*UAS-dlish-RNAi* (#46726), used for all experiments unless otherwise stated.

### From (*Matakatsu and Blair, 2008*, *2012*; Matakatsu, Blair and Fehon, Unpublished)
*hh-gal4 ban3-GFP/TM6B*
*fat$^{fd}$; hh-gal4 ban3-GFP/SM6-TM6B*
*fat$^{fd}$ FRT$^{40A}$/SM6-TM6B*
*fat$^{G-rv}$ FRT40A/SM6-TM6B*
*fat$^{G-rv}$; UAS-ftΔECDΔ?*

*d*$^{GC13}$ *FRT*$^{40A}$/*SM6-TM6B*
*UAS-dachs-V5*
*app*$^{e6}$ *FRT*$^{2A}$/*TM6B*
*app*$^{12-3}$ *FRT*$^{2A}$/*TM6B*
*app*$^{e1}$*FRT*$^{2A}$/*TM6B*
*fat*$^{fd}$ *FRT*$^{40A}$; *app*$^{12-3}$ *FRT*$^{2A}$/*SM6-TM6B*
*hsFLP*; *ft*$^{fd}$ *FRT*$^{40A}$; *ubi-GFP FRT*$^{2A}$/*SM6-TM6B*

## Made by Blair and Fehon labs for this study

*fat*$^{fd}$; *UAS-fat∆ECD-PH+Hippo* (traditional P-element insertion)
*fat*$^{fd}$; *UAS-fat∆ECD-Hippo* (traditional P-element insertion)
*fat*$^{G-rv}$; *UAS-dlish-RNAi (VDRC #46726)*/*SM6-TM6B*
*fat*$^{fd}$ *dlish*$^{04}$/*CyO*
*dachs*$^{GC13}$; *UAS-dachs-∆?-V5* (traditional P-element insertions)
*UAS-dachs-CAAX-HA* (phiC31-mediated integration at *PBac[y*$^+$*-attP-9A]VK00027*)
*UAS-dlish* (phiC31-mediated integration at *PBac[y*$^+$*-attP-9A]VK00027*)
*UAS-dlish-FLAG* (phiC31-mediated integration at *PBac[y*$^+$*-attP-9A]VK00027*)
*dlish*$^{4506}$ (CRISPR, see below)
*dlish*$^{4506}$; *hh-gal4 UAS-ban3-GFP*/*SM5a-TM6B*
*dlish*$^{4506}$; *UAS-dachs-V5* or *UAS-dachs-CAAX-HA*/*SM5a-TM6B*
To induce hsFLP expression, larvae were treated in a 37°C water bath for 90 min.

## Generation of *dlish*$^{4506}$ with the CRISPR-Cas9 system

The guide RNA sequence TTCTTTGCCCCGTGCGCATG, predicted to target near the 5' end of the *dlish* open reading frame, was chosen with fly CRISPR Optical Target Finder (http://tools.flycrispr. molbio.wisc.edu/targetFinder/), and cloned into pCFD3 plasmid. Transgenic guide RNA flies were crossed to *cas9*-expressing flies for one generation, individual G1 lines were crossed to *dlish*$^{04}$, and progeny were screened for PCP and crossvein phenotypes. *dlish*$^{4506}$ deletes nucleotides 17,723,434–17,723,971 on chromosome 2, beginning just 5' to the *dlish* open reading frame and removing coding for amino acids 1–62.

## Quantification of wing sizes

Wings from adult females for each genotype were mounted on glass slides in mineral oil, and images were measured using NIH ImageJ.

## DNA constructs

These were built using standard PCR or the In-Fusion HD Cloning Plus kit (Clontech 638909). pUAST-attB-Dlish-FLAG was amplified from Y2H Mate & Plate Library (Clontech 630485) or clone RE56202 (Drosophila Genomics Resource Center). pUAST-Slimb-FLAG, pUAST-SkpA-FLAG, and pUAST-Cul1-FLAG, were generated from an S2R+ cell cDNA library.

## anti-Dlish serum production and affinity-purification

The full length *dlish* cDNA was amplified from pUAST-attB-Dlish-FLAG, in-fusion cloned into pET-28b(+) (Invitrogen), transformed into BL21(DE3)pLysS *E. coli* competent cells (Promega L1195), and protein expressed induced with 100 µM IPTG at 25°C for 13 hr. His-Dlish protein was purified with HisPur Ni-NTA Superflow Agarose (Thermo 25214), extracted, run on SDS-PAGE, and Coomassie stained. 4–5 mg of His-Dlish protein on the gel was used by Genemed Synthesis Inc. to immunize rabbits. Antiserum was affinity purified using GST-Dlish-coupled Sepharose.

## Immunostaining

Immunostaining of imaginal discs was performed as previously described (*Blair, 2000*), except that for anti-Dachs and anti-Dlish staining fixations were in PBS or Brower's buffer with 4% formaldehyde for 20 min on ice, and in some cases incubations included 5% normal donkey serum. The following primary antibodies were used: rabbit anti-Dlish (1:200), rat anti-Dachs-N1 (1:2,000–10,000) or rat anti-Dachs-C1 (1:10,000) (Matakatsu, Blair and Fehon, Unpublished), rat anti-Fat (1:500, kindly

provided by H. McNeill), rat anti-Ci (1:20) (kindly provided by R. Holmgren) (DSHB Cat# 2A1 Lot# RRID:AB_2109711), mouse anti-En/Inv 4F11 (1:100) (*Patel et al., 1989*; kindly provided by N. Patel), guinea pig anti-Expanded (1:2,000) (*Maitra et al., 2006*), goat anti-DE-Cadherin (1:100) (Santa Cruz Biotechnology Cat# sc-15751 Lot# RRID:AB_639678), rabbit anti-V5 (1:500) (Bethyl Cat# A190-120A Lot# RRID:AB_67586), rabbit anti-FLAG (1:500) (Sigma-Aldrich Cat# F7425 Lot# RRID:AB_439687), and mouse anti-HA (1:500) (Covance Research Products Inc Cat# A594-101L-100 Lot# RRID:AB_291230). Low cross-reactivity secondary antisera were from Jackson ImmunoResearch.

Photos were taken using laser scanning confocal microscopes. To compensate for cell structures lying in different focal planes, most subapical images shown used NIH ImageJ to make Z-projections from individual focal planes that encompass the entire subapical region for all the cells shown.

## Cell culture

S2R+ (DGRC Cat# 196, RRID:CVCL_0A58) and CL8 (DGRC Cat# 151, RRID:CVCL_Z790) cells were cultured at 27°C in Shields and Sang M3 Insect Medium (Sigma) for S2R+ cells or Schneider's Medium (Gibco) for CL8 cells, both supplemented with 10% heat-inactivated fetal bovine serum (Atlanta Biologicals) and 1% penicillin-streptomycin (Gibco). For western blotting or co-IP, cells were seeded in six-well plates for 12 hr, transfected using Effectene (Qiagen 301427) according to the manufacturer's instructions, and harvested after an additional 72 hr.

Plasmids used in vitro were: pUAST-attB-V5-Dlish, pUAST-attB-Dlish-HA, pUAST-Dachs-FLAG, pUAST-Dachs-HA, pCMX-FLAG-Dachs, pUAST-FtΔECD-FLAG, pUAST-FtΔECD-?-HA, pCMX-FLAG-Dlish, pUAST-attB-FLAG-GFP, pGEX2T-Dlish, pGEX2T-Dachs (1110-1232aa), pET-28b(+)-Dlish, pUAST-attB-Dlish, pUAST-Dlish, paw-Gal4, pUAST-attB-Dlish, pUAST-FLAG-Hpo (*Huang et al., 2013*), pUAST-attB-Dco-FLAG, pUAST-attB-DsICD-FLAG, pUAST-3FLAG-Ajuba, pUAST-attB-App-FLAG, pUAST-attB-Fbxl7-FLAG, pUAS-FLAG-Fbxl7 attB (*Bosch et al., 2014*), pUAST-HA-Ubi (*Lee et al., 2012*).

## Immunoprecipitation and western blotting

For co-IP, harvested cells were lysed in 300 μl RIPA lysis buffer (50 mM Tris pH 7.5, 150 mM NaCl, 1 mM EDTA, 1% NP40, 5% glycerol, protease inhibitor (Roche), benzonase, DTT). For S2R+ cells 280 μl of the lysate supernatant was incubated at 4°C first with 1 μg antibodies for 1 hr and next with washed IPA300 Immobilized Protein A (Repligen 10-1003-01) beads for 2 hr; the beads were washed, then treated with 2x sample buffer containing 0.1 M DTT and heated to 99°C for 5 min. For CL8 cells we instead used anti-HA-tag anti-DDDDK-tag magnetic beads (MBL), following the manufacturer's protocol. After SDS-PAGE, proteins were transferred to PVDF membrane (Millipore IPVH00010) which was blocked with 5% milk-PBST for 30 min, incubated in primary antibody at 2 hr at room temperature or overnight at 4°C, washed with PBST, incubated with secondary antibody at room temperature for 1 hr, and washed. Protein was detected either with the ECL kit (Thermo) and exposed to autoradiography film, or using the Li-COR system.

To quantify in vivo protein levels, 10–30 wing discs of each genotype were isolated from late 3rd instar larvae and homogenized with sonication in 2x SDS-PAGE loading buffer and then boiled for 5 min before western blotting as above.

Antisera for western blotting were: Rat anti-Dachs-N1 (1:5,000), rabbit anti-Dlish (1:7,000), mouse anti-α-Tubulin (1:40,000) (Sigma-Aldrich Cat# T5168 Lot# RRID:AB_477579), rabbit anti-FLAG (1:7,000) (Sigma-Aldrich Cat# F7425 Lot# RRID:AB_439687), mouse anti-FLAG (1:6,000) (Sigma-Aldrich Cat# F3165 Lot# RRID:AB_259529), Rabbit anti-V5 (1:5,000) (Bethyl Cat# A190-120A Lot# RRID:AB_67586), Rabbit anti-HA (1:6,000) (Santa Cruz Biotechnology Cat# sc-805 Lot# RRID:AB_631618), mouse anti-HA (1:7,000) (Covance Research Products Inc Cat# BIOT-101L-100 Lot# RRID:AB_10063402), mouse anti-GST (1:6,000) (Santa Cruz Biotechnology Cat# sc-138 Lot# RRID:AB_627677), goat anti-rat-HRP (1:10,000) (Thermo Fisher Scientific Cat# 31,471 Lot# RRID:AB_10965062), goat-anti-rabbit-HRP (1:15,000) (Promega Cat# W4011 Lot# RRID:AB_430833), goat–anti-mouse-HRP (1:10,000) (Jackson ImmunoResearch Labs Cat# 115-035-174 Lot# RRID:AB_2338512), peroxidase-goat anti-mouse IgG-light chain specific (1:10,000) (Jackson ImmunoResearch Labs Cat# 115-035-174 Lot# RRID:AB_2338512), goat anti-rat 680RD (1:10.000) (LI-COR Biosciences Cat# 926–68076 Lot# RRID:AB_10956590), goat anti-mouse 800CW (LI-COR Biosciences Cat# 926–

32210 Lot# RRID:AB_621842) (1:10,000), goat anti-rabbit 680RD (1:10,000) (LI-COR Biosciences Cat# 926–68073 Lot# RRID:AB_10954442).

## Binding between GST-purified and in vitro translated protein

FLAG tagged Dachs and Dlish fragment proteins were generated using Non-Radioactive TNT T7 Quick Coupled Transcription/Translation systems (Promega L1170). 300 µl of binding buffer (1X PBS, 0.1% NP40, 10% Glycerol, 0.5 mM DTT, protease inhibitor) PI, DTT, and 10 µl of the FLAG-tagged proteins were added to cleaned FLAG M2 beads (Sigma A2220), incubated at 4°C for 4 hr, and washed. pGEX2T-Dlish or -Dachs-1110-1232 was expressed in BL21(DE3)pLysS E. coli competent cells (Promega L1195), purified using their GST tags, and 4 µg incubated with protein-bound FLAG beads in binding buffer at 4°C for 2 hr, then washed and boiled for 8 min with 25 µl 2 × SDS/protein sample buffer prior to loading for SDS-PAGE.

## GST pull-down

GST-tagged Dlish domain proteins were prepared as above. App-FLAG or Dachs-FLAG was expressed in S2R+ cells, immunoprecipitated from lysate with FLAG M2 beads (Sigma A2220), and eluted from the beads with 5×FLAG peptides (Sigma F3290). 4 µg purified GST-Dlish domain protein was mixed with 0.5 µg App-FLAG or Dachs-FLAG in binding buffer (1X PBS, 0.1% NP40, 10% glycerol, 0.5 mM DTT, protease inhibitor) with PI and DTT and incubated at 4°C for 2 hr. 20 µl washed GST beads (GE Healthcare 17-0756-01) was added and incubated for an additional 1 hr, washed and then boiled for 8 min with 25 µl 2× SDS/protein sample buffer prior to loading for SDS-PAGE, Coomassie staining and western blotting.

## Palmitoylation assay in S2 cells

The Acyl-Biotin Exchange (ABE) assay of immunoprecipitated proteins was modified from *Brigidi and Bamji (2013)*. Transfected S2R+ cells were lysed in RIPA buffer with protease inhibitor and N-ethylmaleimide (Sigma), incubated for 1 hr at 4°C, and centrifuged. Supernatant was incubated at 4°C with 1 µg antibodies for 1 hr and further incubated with washed IPA300 Immobilized Protein A (Repligen 10-1003-01) overnight. 1/3rd of the beads were treated with RIPA buffer (-HAM control) and the remaining 2/3rds are treated with hydroxylamine hydrochloride (Sigma) (+HAM) at room temperature for 1 hr. The proteins were labeled with EZ-Link BMCC-biotin (Thermo 21900) for 45 min at 4°C, washed with RIPA buffer without NP40 and heated to 99°C for 7 min. Proteins were run on SDS-PAGE, transferred to nitrocellulose membrane and their biotinylation detected with HRP-conjugated streptavidin (1:20,000) (Thermo N100).

## Yeast two-hybrid screen

The yeast two-hybrid screen was performed using the Matchmaker Gold Y2H system (Clontech) according to manufacturer's instructions. Three Dachs baits were used, using the C-terminal 185 (from $G^{1048}NGG...$), 133 (from $P^{1100}RLS...$) or 128 (from $K^{1105}TTS...$) amino acids (numbering from Dachs-PA [*Mao et al., 2006*]).

## Acknowledgements

Thanks to the Bloomington Drosophila Stock Center, the Vienna Drosophila Resource Center, and Drosophila researchers for fly stocks and reagents used in this study.

## Additional information

### Funding

| Funder | Grant reference number | Author |
| --- | --- | --- |
| National Institutes of Health | R01 NS028202 | Seth S. Blair |
| University of Wisconsin-Madison | Department of Zoology Guyer Fellowship | Xing Wang |
| National Institutes of Health | R01 NS034783 | Richard Fehon |

The funders had no role in study design, data collection and interpretation, or the decision to submit the work for publication.

## Author contributions

YZ, XW, HM, SSB, Conception and design, Acquisition of data, Analysis and interpretation of data, Drafting or revising the article; RF, Conception and design, Drafting or revising the article

## Author ORCIDs

Seth S Blair, http://orcid.org/0000-0002-5857-4408

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
