## [Decision Letter]

Thank you for submitting your article "The novel SH3 domain protein Dlish/CG10933 mediates Fat signaling in *Drosophila* by binding and regulating Dachs" for consideration by *eLife*. Your article has been reviewed by two peer reviewers, and the evaluation has been overseen by a Reviewing Editor and K VijayRaghavan as the Senior Editor. The following individual involved in review of your submission has agreed to reveal his identity: Barry Thompson (Reviewer #2).

The reviewers have discussed the reviews with one another and the Reviewing Editor has drafted this decision to help you prepare a revised submission.

Summary:

In this manuscript the authors identify and characterize the SH3 domain protein Dlish as a novel player in the Fat-Dachs pathway upstream of Hippo signaling. Dlish seems to function at the level of Dachs, and like Dachs positively regulate growth. The authors convincingly show that Dlish can interact and colocalize with Dachs, that Dlish and Dachs reciprocally regulate each other's subapical localization and that the localization of both Dlish and Dachs is regulated by Fat in a similar fashion. Strikingly, loss of Dlish can rescue fat mutant flies to adulthood, indicating Dlish plays a major role in Fat signaling and the fat mutant phenoytpe. Additionally, Dlish interacts with and is palmitoylated by App, which is hypothesized to help tether Dachs to the membrane and offers an attractive model for the poorly understood App-Dachs relationship. However, other topics investigated, such as the role of Dlish in regulating Dachs stability or the function of Dlish as a biochemical link between Fat and Dachs, are not conclusively demonstrated and should be investigated more carefully (or removed, as there is already a great deal of data). Overall, the findings are novel and of interest to the field and are appropriate for publication in *eLife* if the following concerns can be addressed.

Essential revisions:

1) Figure 4—figure supplement 1 and Figure 8 show binding assays of different Fat constructs with Dlish and App, respectively. Strangely, the same results are observed for both proteins that don't seem to be conclusive, a fact that is not discussed in the text: in both experiments the "PHHippo" construct strongly binds, while the "Hippo" construct alone doesn't; However, the "∆4-C" construct, which contains the PH domain, doesn't bind, ruling out that it is simply the PH domain that mediates binding. Adding to the confusion, the C-terminus (as tested by ∆N-6) is independently able to interact with Dlish/App. I cannot see how these experiments help the story; I assume that the idea was to link the "Hippo" domain of Fat to Dlish/App and to link back to the regulation of Dachs as shown in Figure 2; but based on the results shown I cannot agree with this logic. Additionally, I would worry that some of these cell culture results might be overexpression artifacts or caused by disrupting the proper structure/folding of Fat.

Considering the data, statements in the text (e.g. at the end of the Results section) are too strong.

2) Figure 4—figure supplement 1 is used to argue that the second SH3 domain of Dlish mediates Dachs binding. A less overexposed blot for Dlish (I assume the IP is shown?) that allows assessment of expression levels, as well as a blot showing Dachs input levels should be provided.

3) Figure 5 would benefit from showing additional panels. For example, in E the reader has to guess where the hh domain starts (please provide a co-stain of Dachs or similar). In F, H and I an additional panel showing a split channel image clearly indicating clone boundaries should be added. Similarly, in Figure 6 panels with Flag stainings need to be shown.

4) Figure 6 It is unclear from text and figure if 100% of fat-fd, *dlish^04^* animals have rescued lethality and survive to adulthood. Please provide information about the efficiency of the rescue.

5) Figure 6 images appear to originate from different discs. It would be more convincing to show the effect of *dlish* knockdown on Dachs localization within the same wing disc and to provide a z-stack (as in Figure 5B2).

6) Figure 6—figure supplement 1 is a poor representation of an adult eye section and should be considered to be removed. If not, a better picture and clarification what "occasional" means ("occasional ommatidial reversal"), should be provided. I is unclear, as the entire panel I1 seems to have purple signal, while I2 suggests that Dachs is represented in purple and gone in the lower third of the panel. The nature of the green staining is not indicated. It is further unclear where the ptc-domain is (provide Dlish co-stain). J is unconvincing, as the Dlish staining looks qualitatively very different from other Dlish stainings shown (for example in K).

7) Figure 6—figure supplement 1 explores the effects of CAAX-mediated membrane tagging of Dachs. At this point, this does not add a lot of value to the manuscript. However, an experiment could be designed to test if the effect on Yki target genes (such as bantam reporter as shown in Figure 6) of Dlish knockdown/*dlish^04^* clones can be repressed/rescued by expression of Dachs-CAAX (e.g. using MARCM). This would argue that the main role of Dlish is indeed to tether Dachs at the membrane. It seems that the authors are trying to implicate something similar in the text (at the end of the subsection “*dlish* is required for normal Dachs localization and fat mutant overgrowth”), however I don't think this conclusion can be easily drawn from overexpression data (levels of dachs-CAAX overexpression would need to be controlled; maybe Dachs-CAAX is less efficient in regulating growth, but the difference in signaling strength is compensated for by increased levels (panel N indicates increased Dachs-CAAX overexpression compared to M).

8) Figure 7 appears to be the result of only one experiment, judging from the lack of error bars. Please provide averaged results from three independent experiments with standard deviation (I realize that one repeat with slightly different numbers is shown in Figure 7—figure supplement 1 but believe three repeats are needed).

9) Figure 7 indicates that Dachs total levels are regulated by Fat and Dlish and suggests this could be due to a recruitment of Slimb, Fbxl7 or other members of the ubiquitination/degradation machinery. However, Figure 7—figure supplement 1 clearly shows that Slimb has no effect on Dachs or Dlish. As this is an important piece of information, please move D-E into the main Figure 7 to not mislead readers.

Figure 7 would greatly benefit from the addition of the ladder (which is generally a good idea), to help the reader evaluate the size of the ubiquitinated band. Please also comment on the fact that despite a reduction in Dachs ubiquitination after Dlish knockdown, total Dachs levels are unchanged, indicating no change in Dachs stability/degradation. And if Dlish is indeed promoting Dachs degradation, why are Dachs levels less increased in *fat-, dlish-RNAi* wing discs than in *fat-* wing discs (panel A-B)?

To strengthen the correlative relationship between Dachs stability/levels and Dlish's ability to bind Slimb and Fbxl7, the authors might consider investigating the ability of Fbxl7 to ubiquitinate/degrade Dlish and to repeat the experiment in Figure 7—figure supplement 1 with *fbxl7* RNAi.

10) Figure 7—figure supplement 1 is very confusing and it is not clear how this adds anything to the story. First of all, this appears to be data from only one experiment, due to the lack of error bars. It is my understanding that S2 cells don't express Fat, which explains why the fat RNAi treatment doesn't have any effect on Dachs, but makes me wonder why this experiment was performed in the first place. Fat∆ECD in S2 cells localizes abnormally, which might explain the opposite effect on Dachs levels than in vivo. At this point the authors have already investigated Dachs level regulation in vivo, which is much more credible. The authors should consider removing this from the manuscript.

11) Figure 8 shows basal/cytoplasmic increase of Dlish and Dachs in app clones. Please provide pictures of the subapical plane or z-axis view of z-stacks. Presumably similar results as in C are expected.

12) The assay used by the authors to argue that Dlish is palmitoylated is not specific enough to make this claim. Any other cysteine modifications sensitive to hydroxylamine would also give a signal – including any other fatty acids. The conclusions should be modified to reflect this uncertainty – alternatively using "clickable' palmitate could give a more interpretable result.

---

## [Author Response]

*In this manuscript the authors identify and characterize the SH3 domain protein Dlish as a novel player in the Fat-Dachs pathway upstream of Hippo signaling. Dlish seems to function at the level of Dachs, and like Dachs positively regulate growth. The authors convincingly show that Dlish can interact and colocalize with Dachs, that Dlish and Dachs reciprocally regulate each other's subapical localization and that the localization of both Dlish and Dachs is regulated by Fat in a similar fashion. Strikingly, loss of Dlish can rescue fat mutant flies to adulthood, indicating Dlish plays a major role in Fat signaling and the fat mutant phenoytpe. Additionally, Dlish interacts with and is palmitoylated by App, which is hypothesized to help tether Dachs to the membrane and offers an attractive model for the poorly understood App-Dachs relationship. However, other topics investigated, such as the role of Dlish in regulating Dachs stability or the function of Dlish as a biochemical link between Fat and Dachs, are not conclusively demonstrated and should be investigated more carefully (or removed, as there is already a great deal of data). Overall, the findings are novel and of interest to the field and are appropriate for publication in eLife if the following concerns can be addressed.*

First, we, like the reviewers, felt that the activity of Dlish and Approximated needed to be tied more closely to the activity of the ICD of Fat. Our co-IP between Dlish, Approximated and the Fat ICD suggested such a connection, and we now have new data that greatly strengthens this. As we showed previously, adding Approximated to our in vitro palmitoylation assay increased the palmitoylation of Dachs. We now show that this App-stimulated palmitoylation is reliably decreased by the addition of Fat△ECD (6 of 6 trials). This supports our model that the Fat ICD inhibits the activity of Approximated, decreasing the palmitoylation of Dlish and therefore the tethering of the Dlish-Dachs complex to the cell membrane. This new data is included at the end of the Results, as part of a new figure that combines the new and old palmitoylation data. This also provides a strong rationale for retaining the co-IP data upon which the model is based.

Next, we generated a new *dlish* mutant that allowed us to compare the activities of overexpressed Dachs and the membrane-directed Dachs-CAAX in a *dlish* mutant background. As predicted, the removal of Dlish inhibits the activity of Dachs, but this requirement is overridden by directing Dachs-CAAX to the cell membrane. We describe this valuable experiment in more detail below in our comments on Essential Revision 7.

Third, we have added multiple trials and statistics to our demonstration of increases in total Dachs levels in *dlish* mutant, *fat* mutant and *fat dlish* double mutants. We have also greatly increased the number of trials for our Dlish palmitoylation assays, and this has allowed us to perform statistical analyses of the effects of App (and now Fat△ECD). Both of these new data sets strengthen the claims we made in the original manuscript.

Finally, we strongly feel that it is worthwhile retaining our data on the regulation of Dachs levels by Dlish and Fat, as it is both novel and a prominent feature of their phenotypes, and constrains the models of how they work. We also have new evidence that *fbxl7* can regulate subapical Dlish in vivo, and with a stronger *slimb* loss than the RNAi we used previously, can now also see increases in subapical Dlish and Dachs. This provides a much stronger rationale for the co-IP binding to Fbxl7 and Slimb.

However, we agree that the data on ubiquitination is incomplete, and we have removed it for use in a future study. As the reviewers pointed out, we have not worked out the basis for our in vitro ubiquitination result, and moreover Dlish could be a target of Fbxl7 and Slimb. The mutual co-dependence of subapical Dlish and Dachs makes this difficult to determine with any in vivo genetic test, as the absence of one will prevent subapical localization of the other, leaving none for subapical Fbxl7 or Slimb to work on. Our first in vitro tests of Dlish ubiquitination have been slowed by difficulties getting a strong enough Dlish signal, and this has prevented us from completing them in the time frame of the revision. We have nonetheless retained some speculation about ubiquitination in the Discussion.

Essential revisions:

*1) Figure 4—figure supplement 1 and Figure 8 show binding assays of different Fat constructs with Dlish and App, respectively. Strangely, the same results are observed for both proteins that don't seem to be conclusive, a fact that is not discussed in the text: in both experiments the "PHHippo" construct strongly binds, while the "Hippo" construct alone doesn't; However, the "∆4-C" construct, which contains the PH domain, doesn't bind, ruling out that it is simply the PH domain that mediates binding. Adding to the confusion, the C-terminus (as tested by ∆N-6) is independently able to interact with Dlish/App. I cannot see how these experiments help the story; I assume that the idea was to link the "Hippo" domain of Fat to Dlish/App and to link back to the regulation of Dachs as shown in Figure 2; but based on the results shown I cannot agree with this logic. Additionally, I would worry that some of these cell culture results might be overexpression artifacts or caused by disrupting the proper structure/folding of Fat.*

*Considering the data, statements in the text (e.g. at the end of the Results section) are too strong.*

As discussed in the text, the C-terminal ∆N-6 region in Fat’s ICD likely plays an important role modulating the activity of the Hippo domains. It also binds to several of the proteins that bind to Dlish by co-IP (App, Fbxl7, Dco), as well as others (Grunge/Atrophin, Lowfat), and the region is phosphorylated by Dco. That Dlish can complex with this region is therefore not so surprising. What is important is that App and Dlish can co-IP with the PHHippo domains. This is the first report of any protein complexing with this critical region, and in our studies co-IP this to this region has been rare. We feel strongly that this needs to be reported.

We do not see binding to the smaller Hippo region in S2 cells, even though the identical region retains some, although weaker, biological activity in vivo. As the reviewers mention, there may be some trivial reasons for this, such as abnormal protein folding in S2 cells. We thought it would be dishonest not to include this data, however. Therefore we have added the following to the text about the Dlish co-IP:

“We did not observe co-IP with the smaller Fat△ECD-Hippo construct that lacks the PH domain. Although in vivo the PH region is not absolutely required for activity of the adjacent Hippo domains (e.g. Figure 2—figure supplement 1; Matakatsu and Blair 2012; Pan et al. 2013), it does strengthen that activity (Matakatsu and Blair 2012), and in the different environment found in S2 cells may be more critical for proper folding or the formation of protein complexes.”

We have also changed the text as follows:

“We have confirmed binding in S2 cells, and found that App can co-IP not only with the domains C-terminal to the Hippo domains (Fat△ECD-△N-6) where biotinylation occurs, but also the domains most strongly active in Dachs and Hippo pathway regulation (Fat△ECD-PHHippo) (Figure 12). Like Dlish, App failed to bind to Fat△ECD△4-C and Fat△ECD-Hippo, which have, respectively, no or reduced Hippo activity in vivo (Matakatsu and Blair, 2012).”

*2) Figure 4—figure supplement 1 is used to argue that the second SH3 domain of Dlish mediates Dachs binding. A less overexposed blot for Dlish (I assume the IP is shown?) that allows assessment of expression levels, as well as a blot showing Dachs input levels should be provided.*

Although we thought the levels were clear on the original, we have as requested substituted a lower exposure, and labeled this more clearly as an IP. Also, as the loading levels for Dlish-△6, containing just the third SH3 of Dlish, are lower than those of the other constructs, we have added an equivalent GST pulldown experiment to Figure 4—figure supplement 2 that shows the same failure in Dachs binding to a GST-Dlish containing just the third SH3 domain, even with high input levels.

We have added back the GST-tagged input lane to all of our IVT experiments. Since the GST-tagged protein is first purified as a single preparation, and then equal amounts of that preparation incubated with each of the FLAG-tagged, in vitro translated binding candidates, there were equal levels of GST-protein input for each of the FLAG-tagged binding candidates. The single input lane is thus a bit redundant, as it confirms what is already obvious from the positive anti-FLAG IPs, that we successfully made and purified GST-tagged protein.

*3) Figure 5 would benefit from showing additional panels. For example, in E the reader has to guess where the hh domain starts (please provide a co-stain of Dachs or similar). In F, H and I an additional panel showing a split channel image clearly indicating clone boundaries should be added. Similarly, in Figure 6 panels with Flag stainings need to be shown.*

We were originally trying to save space in what were already quite large figures. We have however added the extra frames for the original 4H, 6N and 6O. For the original 4F and 4I we have increased the intensity of the clone marker in the multicolor frames to make the clone regions obvious, and added an outline to the original 4I. And as some might question whether the clone in 4I is in the correct focal plane, being near a fold in the disc, we added a control frame showing unchanged subapical DE-cadherin from the same focal plane. To make room, we removed the adjacent RFP clone marker panel from the *fat dachs* double mutant clone, as there was already a panel marking the clone boundary by the absence of Dachs.

With these and other requested changes the original Figure 5 (now Figure 6) has gotten a bit large, so we have removed the *fat* mutant data and made it into a separate figure.

*4) Figure 6 It is unclear from text and figure if 100% of fat-fd, dlish^04^ animals have rescued lethality and survive to adulthood. Please provide information about the efficiency of the rescue.*

The rescue is over 50%; we have added the phrase “the majority” to the text and added precise numbers to the figure legend. Keeping in mind that this only rescues the Hippo signaling aspect of Fat function, and not the orientation information for planar cell polarity that would be transmitted by binding between the extracellular domains of Fat and Dachsous, this level of rescue is quite good, better than we get by expressing Fat△ECD and about the same as we get expressing full-length Fat in *fat* mutants (Matakatsu and Blair, 2012).

*5) Figure 6 images appear to originate from different discs. It would be more convincing to show the effect of dlish knockdown on Dachs localization within the same wing disc and to provide a z-stack (as in Figure 5B2).*

We have added a cross-section Z-stack image of a single disc, with the posterior labeled with anti-Engrailed. This is in fact anti-Dachs and anti-En from the same Z-stack used to show Dlish localization in the original Figure 5B2.

*6) Figure 6—figure supplement 1 is a poor representation of an adult eye section and should be considered to be removed. If not, a better picture and clarification what "occasional" means ("occasional ommatidial reversal"), should be provided.*

We have added a lower magnification image of the retina, and a larger detail with the polarities marked, so that an example of a reversed ommatidium can be more easily compared with its normal neighbors. Reversals were rare, around 2%, and we have added that information to the text.

*I is unclear, as the entire panel I1 seems to have purple signal, while I2 suggests that Dachs is represented in purple and gone in the lower third of the panel. The nature of the green staining is not indicated. It is further unclear where the ptc-domain is (provide Dlish co-stain).*

The green staining was in fact indicated as being *ptc-gal4*-driven *UAS-GFP* (“*ptc > GFP*”). Thus it shows the anterior *ptc-gal4* domain, so Dlish staining is not needed. We agree, however, that the green hides the higher basolateral Dachs shown in purple in the double color image, so we have simply followed the reviewer’s previous suggestion and now show GFP and Dachs in side-by-side panels.

*J is unconvincing, as the Dlish staining looks qualitatively very different from other Dlish stainings shown (for example in K).*

We have changed this to a better image, and for reasons of continuity now added this to the main figures (see below). The reason for the difference in the original image is that subapical anti-Dlish staining is never as clean in the *dlish^04^*/ heterozygotic discs as they are in wild type discs, likely due to low levels or abnormal localization of the Dlish^04^ protein.

*7) Figure 6—figure supplement 1 explores the effects of CAAX-mediated membrane tagging of Dachs. At this point, this does not add a lot of value to the manuscript. However, an experiment could be designed to test if the effect on Yki target genes (such as bantam reporter as shown in Figure 6) of Dlish knockdown/dlish^04^ clones can be repressed/rescued by expression of Dachs-CAAX (e.g. using MARCM). This would argue that the main role of Dlish is indeed to tether Dachs at the membrane. It seems that the authors are trying to implicate something similar in the text (at the end of the subsection “dlish is required for normal Dachs localization and fat mutant overgrowth”), however I don't think this conclusion can be easily drawn from overexpression data (levels of dachs-CAAX overexpression would need to be controlled; maybe Dachs-CAAX is less efficient in regulating growth, but the difference in signaling strength is compensated for by increased levels (panel N indicates increased Dachs-CAAX overexpression compared to M).*

We agree that the cleaner experiment is not to rely on the opposite effects of Dachs and Dachs-CAAX expression on Dlish levels, as in the original manuscript, but rather see how the different Dachs constructs act in a *dlish* mutant background. We were not previously able to do this experiment because of the UAST-YFP.Rab26.Q250L P element insertion of the *dlish^04^* allele, which causes wing abnormalities in the presence of the Gal4 used to drive *UAS-dachs* expression. However, as the manuscript was out for review we obtained a new CRISPR allele of *dlish* that gives phenotypes very similar to *dlish^04^* (now shown in the revised manuscript). The experiment required building new stocks with this allele, which occupied much of our time since we received the reviews. Importantly, the results match our prediction: the *dlish* mutation greatly reduced the overgrowth and Yki target gene expression induced by *UAS-dachs*, but not those induced by *UAS-dachs-CAAX.* We also show that overexpressed Dachs-CAAX remains cortically localized in the *dlish* mutant, while Dachs becomes largely cytoplasmic. Therefore, membrane localization of Dachs can compensate for the loss of Dlish. We have substituted this new experiment for our previous results, and moved the data from the supplement to a new figure.

*8) Figure 7 appears to be the result of only one experiment, judging from the lack of error bars. Please provide averaged results from three independent experiments with standard deviation (I realize that one repeat with slightly different numbers is shown in Figure 7—figure supplement 1 but believe three repeats are needed).*

While each lane in each trial contains extracts from multiple wing discs, we agree that there is some variation from trial to trial due to the vagaries of western blotting. We have therefore as requested repeated this experiment multiple times. Our largest number of trials were using *dlish^04^*, and we have put this data and the statistics into the main figure; the results are significant using both non-parametric and parametric tests. Our trials using *dlish-RNAi* are still too few to do the same type of statistical analysis, but give the same general result, so we have included these in the supplemental figure.

*9) Figure 7 indicates that Dachs total levels are regulated by Fat and Dlish and suggests this could be due to a recruitment of Slimb, Fbxl7 or other members of the ubiquitination/degradation machinery. However, Figure 7—figure supplement 1 clearly shows that Slimb has no effect on Dachs or Dlish. As this is an important piece of information, please move D-E into the main Figure 7 to not mislead readers.*

We have improved this experiment by using the stronger loss-of-function *slimb* mutant clones, and now see slight increases in subapical Dachs and Dlish. We include this in a revised main figure. As the previous result was mentioned prominently in the main text, there was certainly no attempt to mislead the reader.

*Figure 7 would greatly benefit from the addition of the ladder (which is generally a good idea), to help the reader evaluate the size of the ubiquitinated band. Please also comment on the fact that despite a reduction in Dachs ubiquitination after Dlish knockdown, total Dachs levels are unchanged, indicating no change in Dachs stability/degradation. And if Dlish is indeed promoting Dachs degradation, why are Dachs levels less increased in fat-, dlish-RNAi wing discs than in fat- wing discs (panel A-B)?*

*To strengthen the correlative relationship between Dachs stability/levels and Dlish's ability to bind Slimb and Fbxl7, the authors might consider investigating the ability of Fbxl7 to ubiquitinate/degrade Dlish and to repeat the experiment in Figure 7—figure supplement 1 with fbxl7 RNAi.*

The remainder of comment 9 concerned the ubiquitination experiment which has been removed from this version of the manuscript.

*10) Figure 7—figure supplement 1 is very confusing and it is not clear how this adds anything to the story. First of all, this appears to be data from only one experiment, due to the lack of error bars. It is my understanding that S2 cells don't express Fat, which explains why the fat RNAi treatment doesn't have any effect on Dachs, but makes me wonder why this experiment was performed in the first place. Fat∆ECD in S2 cells localizes abnormally, which might explain the opposite effect on Dachs levels than* in vivo*. At this point the authors have already investigated Dachs level regulation* in vivo*, which is much more credible. The authors should consider removing this from the manuscript.*

Comment 10 concerned the ubiquitination experiment that has been removed from this version of the manuscript.

*11) Figure 8 shows basal/cytoplasmic increase of Dlish and Dachs in app clones. Please provide pictures of the subapical plane or z-axis view of z-stacks. Presumably similar results as in C are expected.*

While the change in cytoplasmic Dlish is quite obvious (and we now include a better example), subapical anti-Dlish stain in our FRT *app^e6^* heterozygote discs is degraded compared with wild type discs, perhaps due to the lower dosage of *app*. While we have tried several trials of this experiment, the subapical signal outside the clone is simply too poor to make this a fair test. So we instead rely on the very convincing reduction in subapical Dlish caused by *app* loss compared with the abnormally high Dlish levels in *fat/fat; app*/+ discs.

*12) The assay used by the authors to argue that Dlish is palmitoylated is not specific enough to make this claim. Any other cysteine modifications sensitive to hydroxylamine would also give a signal – including any other fatty acids. The conclusions should be modified to reflect this uncertainty – alternatively using "clickable' palmitate could give a more interpretable result.*

The reviewer makes a good point about the ABE possibly detecting other cysteine modifications, and so we have changed the text about our results from “palmitoylation” to the more accurate “HAM-induced biotinylation”. However, the sensitivity of this biotinylation to the levels of the palmitoyltransferase App make it very unlikely that change is due to some other modification. We have therefore also added the following text:

“This strongly argues that a substantial fraction of the biotinylation represents App-sensitive palmitoylation.”